# A Rich Knowledge Space for Scalable Deepfake Detection

**Inho Jung  Hyeongjun Choi  Binh M. Le  Hohyun Na  Simon S. Woo**[*]
College of Computing and Informatics
Sungkyunkwan University, Suwon, Republic of Korea
{inhovation97, junhjun, bmle, skghgus9, swoo}@g.skku.edu

## Abstract

The proliferation of realistic deepfakes has driven the development of numerous benchmark datasets to support detection research. Despite their increasing volume and diversity, no prior effort has systematically consolidated these resources into a unified framework for large-scale model training, nor has there been a massively pre-trained model tailored to deepfake detection. In this work, we introduce **M**ulti-modal **M**ulti-type **I**ntegrated **D**eepfake **D**ataset (**MMI-DD**), a large-scale resource containing 3.6 million facial images, the largest collection to date. It unifies diverse benchmarks with uniform preprocessing, and further provides fine-grained annotations across four deepfake types, as well as VLM-generated descriptions capturing both facial and environmental attributes for each image. By leveraging this comprehensive multi-modal dataset, we construct a foundational deepfake knowledge space that empowers our model to discern a broad spectrum of synthetic media. Our method, $\mathcal{SD}^2$ (Scalable Deepfake Detection), refines CLIP for deepfake detection, optimizing image-text classification with rich, type-specific labels. We enhance this with intermediate visual features capturing low-level cues and text label separation loss for stability. We further leverage VLM-generated descriptions and contrastive learning to expand the scope of forgery knowledge, reducing overfitting and enhancing generalization. Extensive experiments on challenging deepfake datasets and AIGC benchmark demonstrate the effectiveness, scalability, and real-world applicability of our approach.

## 1 Introduction

Over the past decade, significant advancements in deep learning have enabled the creation of highly realistic synthetic media, often misused for malicious purposes (Patrini, 2019; Donie, 2019; Rachel & Donie, 2019; Romano, 2019). The proliferation of user-friendly, open-access deepfake tools (DeepFaceLab, 2023; FaceSwap, 2016; Siarohin et al., 2019) has further exacerbated and accelerated the rapid spread of such content across the digital landscape. This trend has raised profound societal concerns regarding its implications for security, privacy, and trust in digital media (Federal Bureau of Investigation (FBI), 2022; Li et al., 2022). Consequently, detecting deepfake media has emerged as a critical challenge in computer vision, necessitating the development of robust solutions supported by large-scale resources (Yan et al., 2024; Dolhansky et al., 2020).

Researchers have actively pursued the development of robust deepfake detection models, spurred by the emergence of deep learning. These methods scrutinize various dimensions of synthesis artifacts, such as spatial inconsistencies (Nguyen et al., 2019; Le & Woo, 2023; Tariq et al., 2021), frequency-domain irregularities (Qian et al., 2020; Song et al., 2022; Le & Woo, 2022), and temporal anomalies (Wang et al., 2023b; Zheng et al., 2021; Gu et al., 2021), and perceptually trace those artifacts (Tan et al., 2023; Lin et al., 2023). In parallel, worldwide research groups have developed forensic datasets to support both the training and evaluation of detectors, encompassing enterprise-led efforts (Dolhansky et al., 2020; Dufour & Gully, 2019; Kwon et al., 2021), as well as academic contributions (Rossler et al., 2019; Cho et al., 2023; Yan et al., 2024). Nevertheless, most prior work (Zhao et al., 2021; Wang et al., 2023b; Zheng et al., 2021; Shiohara & Yamasaki, 2022; Ni

---

[*]Corresponding author

et al., 2022; Cao et al., 2022; Dong et al., 2023; Dat et al., 2024) adheres to conventional practices, training on a single dataset (*e.g.,* FaceForensics++) and validating on others (*e.g.,* DFDC). However, this training paradigm poses three fundamental limitations: (i) the model sees only a narrow range of forensic artifacts, restricting its ability to detect more advanced manipulations; (ii) it becomes biased toward the source dataset, reducing robustness on unseen data with varying demographic groups, capture conditions, or post-processing; and (iii) a single-dataset training setup conceals the model's limitations in heterogeneous contexts and misleads about its practical applicability where training data naturally arise from multiple sources. Hence, there is an urgent need to leverage the full spectrum of available resources to develop detectors that enhance both robustness and knowledge, establishing practical benchmarks and applicability in real-world scenarios.

Nevertheless, training models on large-scale, heterogeneous datasets remains challenging due to distributional shifts and the risk of poor generalization. To explore this, we investigate three CLIP (Radford et al., 2021)-adapted baselines commonly used for deepfake detection as illustrated in Fig. 1 (left). Each model is progressively trained on our large-scale dataset (Sec. 3), scaling from 100K to 3M images sourced from 11 distinct datasets. Models are then evaluated on cross-domain benchmarks (see Sec. 5.2.1). Surprisingly, even with these state-of-the-art (SOTA) CLIP adaptation methods, we observe a notable performance degradation as training data size increases. This finding highlights the difficulty of maintaining generalization across massive, diverse data, and motivates our proposed approach. Our method directly addresses these limitations through multi-modal supervision with carefully designed training objectives.

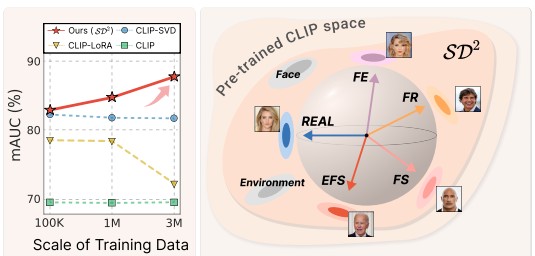

Figure 1: Scalability and goal of our approach. (Left) As training data grows, baseline detectors[1] reach limits and saturate, unable to exploit larger datasets, while our $\mathcal{SD}^2$ steadily improves detection performance, overcoming these constraints. (Right) $\mathcal{SD}^2$ aims to forge a foundational deepfake knowledge space that effectively leverages large-scale, diverse datasets.

In this research, we propose a novel CLIP-based visual-language learning strategy for deepfake detection that unifies representations across diverse deepfake datasets with varying distributions, dubbed $\mathcal{SD}^2$ (Scalable Deepfake Detection). Our approach consolidates multiple datasets, incorporating a broad spectrum of forgery knowledge. To this end, we construct **M**ulti-modal **M**ulti-type **I**ntegrated **D**eepfake **D**ataset (**MMI-DD**) by integrating a broad range of deepfake sources. Our dataset has three key features: (i) it incorporates diverse sources and manipulation types with uniform preprocessing; (ii) each image is meticulously annotated into one of five types; and (iii) each image is paired with VLM-based text descriptions capturing facial attributes and environmental context, resulting in a large-scale multi-modal dataset for unified visual-language learning. Leveraging this dataset, our model builds a comprehensive knowledge base (Fig. 1 (right)).

We first propose **Cross-Layer Attention Module (CLAM)**, which fuses visual features from the final and intermediate layers of CLIP. It captures multi-level forgery cues to support our learning strategies: classification and contrastive learning. For classification, we leverage type-specific *text labels* matched to images, enabled by our detailed annotations. Using these labels, we introduce our **Fine-Grained Image-Text Classification Loss** that distinguishes real from four forgery types in the embedding space, extending the detector's knowledge beyond a binary decision. Also, we add **Text Label Separation Loss** to stabilize training by explicitly separating text label embeddings associated with each type. Moreover, we propose **Dual Image-Text Contrastive Loss**, which aligns each image with two types of *VLM-generated descriptions* to enhance semantic-visual coherence. This objective mitigates overfitting to dataset-specific artifacts arising from solely relying on cross-entropy loss, as observed by Wang et al. (2024b) and Sun et al. (2023).

Finally, we conduct extensive experiments to validate our $\mathcal{SD}^2$ on intra- and cross-domain datasets from popular benchmarks. We show that our model accommodates a broader range of training data

---

[1]All baseline detectors are based on the CLIP architecture, including the state-of-the-art CLIP-SVD (Effort) (Yan et al., 2025b). Detailed descriptions and extensive analysis are provided in Sec. 5.3.

and achieves superior generalization performance on challenging real-world datasets. Additionally, we demonstrate that the CLIP image encoder, fine-tuned with our strategies, achieves SOTA results on non-facial AI-Generated Content (AIGC) benchmark, highlighting its applicability.

Our main contributions are summarized as follows: (i) we introduce **MMI-DD**, a multi-modal integrated dataset by collecting deepfake datasets annotated with five categories, along with two types of text descriptions, enabling effective utilization of a broad range of resources for training. By releasing **MMI-DD**, we provide the community with a large-scale, multi-modal resource, fostering broader exploration and study in deepfake detection; (ii) we propose $\mathcal{SD}^2$, a novel visual-language learning framework that not only captures rich semantic attributes through textual information but also demonstrates unprecedented scalability, consistently improving with data volume, unlike prior methods that saturate or degrade; and (iii) we demonstrate that our approach outperforms existing SOTA baselines on both real-world deepfake benchmarks and AIGC detection benchmark.

## 2 RELATED WORK

### 2.1 CONVENTIONAL DEEPFAKE DETECTION

With the advent of deep learning, CNN-based models (He et al., 2016; Chollet, 2017; Tan & Le, 2019) and Transformer-based architectures (Vaswani et al., 2017) have dominated the field of face forensics detection, leveraging large-scale datasets to enhance classification performance. Many approaches improve generalization to unseen benchmark datasets through architectural innovations (Rossler et al., 2019; Zhao et al., 2021; Zheng et al., 2021) and augmentation strategies (Shiohara & Yamasaki, 2022; Dat et al., 2024). However, most studies (Zhao et al., 2021; Wang et al., 2023b; Zheng et al., 2021; Shiohara & Yamasaki, 2022; Ni et al., 2022; Cao et al., 2022; Dong et al., 2023; Dat et al., 2024) adhere to a conventional paradigm, training on a single dataset and validating on others. While methodologically advanced, this approach limits the diversity of knowledge models can acquire and their practical utility due to constrained training data. To address this challenge, several works have expanded training data volume (Lai et al., 2024; Amin et al., 2023) by integrating multiple benchmark datasets, yet these efforts remain limited by the scope and modality of available deepfake datasets.

### 2.2 DEEPFAKE DETECTION VIA MULTI-MODAL MODELS

The rapid progress of Vision-Language Models (VLMs) has demonstrated impressive visual understanding and natural language interaction capabilities. Particularly, CLIP (Radford et al., 2021) has emerged as a promising foundation model for deepfake detection (Ojha et al., 2023), which leverages its frozen image embeddings for classification. Subsequent studies (Baraldi et al., 2024; Koutlis & Papadopoulos, 2024; Yan et al., 2025a; Cui et al., 2024) also utilize the frozen CLIP encoder, focusing on extracting fine-grained forgery cues suitable for deepfake detection tasks. Recent research has expanded the use of CLIP to improve generalization by incorporating not only its visual part but also its textual part (Wu et al., 2023; Khan & Dang, 2024). Further extending beyond simple text labels, natural language is incorporated to provide detailed supervision for CLIP fine-tuning (Sun et al., 2023), and class-aware deepfake classification objectives are introduced through prompt design (Wang et al., 2024b). Despite these advancements, most studies still adhere to the traditional single-dataset training paradigm, failing to fully exploit the scalability of pre-trained models from a data-centric perspective. In this work, we propose a methodology that integrates heterogeneous training datasets, harnessing the capacity of CLIP for generalizable deepfake detection.

## 3 LARGE-SCALE DEEPFAKE DETECTION DATASET

### 3.1 DATASET COLLECTION

We collect and preprocess diverse deepfake benchmark datasets to facilitate joint training across multiple datasets, which are summarized in Tab. 1. The recently introduced DF40 (Yan et al., 2024), which is known for its extensive data diversity, has primarily been used for evaluation. In contrast, we are the first to leverage the entire training set from DF40 for model training, integrating them with six additional popular deepfake datasets: DFF (Song et al., 2023), DFFD (Dang et al., 2019),

Table 1: Summary of our integrated dataset, MMI-DD, including the numbers of real and fake images and their type annotations.

| Dataset | Forgery Types | Real Images | Fake Images | Total |
|---|---|---|---|---|
| DF40 | *FS, FR, EFS, FE* | 77,200 | 1,613,000 | 1,690,200 |
| DFF | *EFS* | 24,200 | 57,500 | 81,700 |
| DFFD | *FE* | 20,000 | 136,000 | 156,000 |
| DFDCP | *FS* | 11,700 | 38,800 | 50,500 |
| KoDF | *FS* | 11,500 | 28,200 | 39,700 |
| FF++ (c40) | *FS, FR* | 32,000 | 160,000 | 192,000 |
| TIMIT (LQ) | *FS* | - | 10,240 | 10,240 |
| TIMIT (HQ) | *FS* | - | 10,240 | 10,240 |
| CelebA | - | 202,600 | - | 202,600 |
| CelebA-HQ | - | 30,000 | - | 30,000 |
| CelebV-HQ | - | 1,051,100 | - | 1,051,100 |
| FFHQ | - | 70,000 | - | 70,000 |
| **Overall** | - | 1,530,300 | 2,053,980 | 3,584,280 |

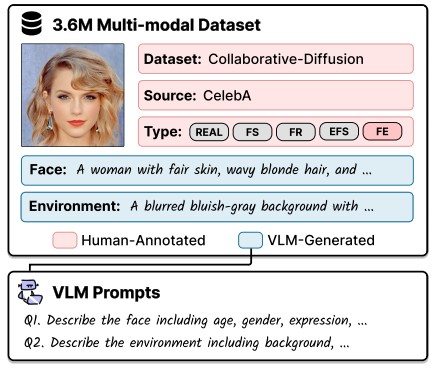

Figure 2: Illustration of detailed annotations in the integrated dataset.

DFDCP (Dolhansky et al., 2020), KoDF (Kwon et al., 2021), FF++ (c40) (Rossler et al., 2019), and TIMIT (Korshunov & Marcel, 2018) (including high-quality (HQ) and low-quality (LQ) variants). To balance the ratio of real and fake images, we further incorporate four additional real datasets: CelebA (Liu et al., 2015), CelebA-HQ (Karras et al., 2017), CelebV-HQ (Zhu et al., 2022), and FFHQ (Karras et al., 2019). This results in a large-scale facial image dataset comprising a total of 3.6 million images. All deepfake datasets are preprocessed in a consistent manner by cropping face regions. To address potential challenges in large-scale dataset integration (*e.g.,* preprocessing inconsistencies, identity overlap, and data leakage), we implement rigorous data partitioning protocols. Detailed descriptions of the datasets, along with our protocols for data cleaning and partitioning, are thoroughly documented in Sec. A of the Appendix.

**Categorical Annotation with Text Augmentation.** Our six researchers independently and manually categorize images from the datasets, and cross-check one another. Following prior works (Yan et al., 2024; Mirsky & Lee, 2021), each image is labeled as either *REAL* or one of four deepfake types: *Face Swapping (FS), Face Reenactment (FR), Entire Face Synthesis (EFS)*, and *Face Editing (FE)*. For each type, we construct a set of *text labels*. During training, text labels corresponding to each image type serve as anchors for fine-grained deepfake classification based on image-text similarity. Unlike previous works (Khan & Dang, 2024; Wu et al., 2023) that use simple text labels such as `"a photo of a [real/fake] image"`, we augment text labels from a simple label to about 30 comprehensive labels using GPT-o1 (OpenAI, 2024) and human feedback. For example, for the *Face Swapping*, we generate a range of diverse labels, including the simple label `"a photo of a Face Swapping"` as well as more detailed labels such as `"a deepfake-crafted photo replacing one individual's face with another's"`. This augmentation encourages the model to develop a deeper semantic understanding of each type and can enhance its ability to distinguish different types of deepfakes. Please visit Sec. E of the Appendix for more details of text annotations and their augmentation.

## 3.2 DATASET ANNOTATION

**Contextual Text Generation.** While traditional categorical labels suffice for training a deepfake classifier, portrait images contain not only forensic cues, but also unrelated details such as demographic traits or background contexts. Recent work by (Wang et al., 2025) empirically and theoretically demonstrates that when spurious features such as background or lighting strongly correlate with the training data's text descriptions, CLIP's contrastive objective aligns image embeddings with these features. Consequently, when repurposed for deepfake detection, this leads the model to focus more on such spurious correlations rather than forensic signals, resulting in a less robust detector. To address this vulnerability, we propose generating comprehensive *text descriptions* for all images to disentangle forensic cues from contextual elements, enhancing the model's understanding beyond binary classification. Our ablation studies (Sec. 5.3) further show that including contextual descriptions improves performance on unseen datasets. To generate these auxiliary descriptions,

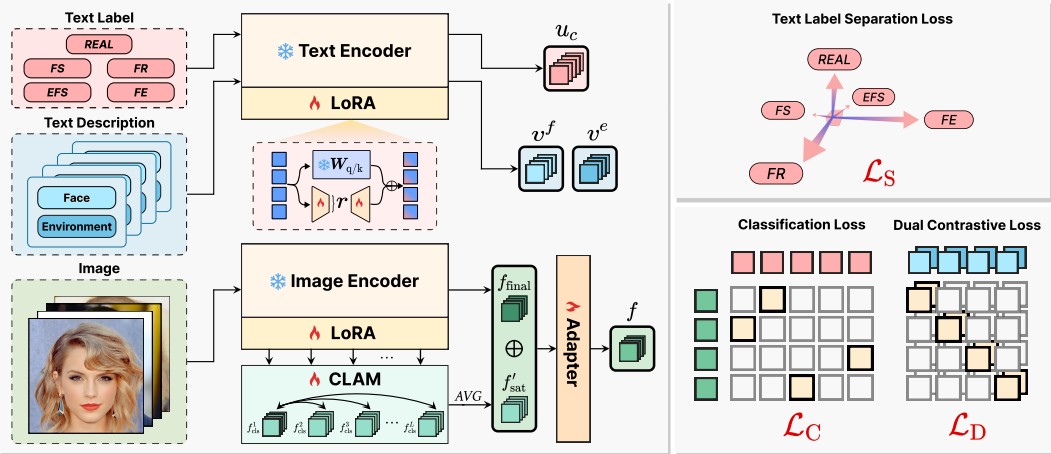

Figure 3: Overview of our $\mathcal{SD}^2$ training framework. $\mathcal{SD}^2$ employs CLIP text and image encoders, both fine-tuned with LoRA. The Cross-Layer Attention Module (CLAM) enhances visual features by fusing low-to-high-level information. The model is optimized with three objectives: Classification Loss ($\mathcal{L}_C$) to distinguish real and four fake types, Text Label Separation Loss ($\mathcal{L}_S$) to enforce separation among types, and Dual Contrastive Loss ($\mathcal{L}_D$) to align image-text pairs.

we leverage InternVL (Chen et al., 2024) in a Visual Question Answering setting. Specifically, we use two distinct prompts to obtain different aspects of the images: (1) *facial* that captures facial attributes and (2) *environmental* that describes background context. To validate the quality of the VLM-generated descriptions, we manually audit randomly sampled images from each dataset and confirm that the descriptions align with the visual content. Furthermore, recent research (Lu & Zhong, 2024) demonstrates that SOTA VLMs exhibit a high degree of perceptual alignment with human observations, particularly in describing human portrait images. This establishes our use of VLM-driven annotation as a reliable and scalable alternative to manual efforts for large-scale dataset construction. Examples of our annotated data are presented in Fig. 7, and further details are provided in Sec. F, both included in the Appendix.

Our dataset annotation procedure, combining human and VLM-generated annotations, is illustrated in Fig. 2. To the best of our knowledge, **MMI-DD** is among the largest and most diverse collections available for training deepfake detection models. It enables the construction of a comprehensive knowledge space and a more diverse foundation for face forensics, in contrast to conventional approaches that train models on a single dataset.

## 4   $\mathcal{SD}^2$: VISUAL-LANGUAGE LEARNING FOR DEEPFAKE DETECTION

We propose a novel multi-modal learning framework for deepfake detection based on CLIP (Radford et al., 2021), which integrates image and text modalities. Figure 3 illustrates our overall proposed training framework. We define a data format $S = \{(x_n, c_n, d_n^f, d_n^e)\}_{n=1}^N$, where $x \in \mathcal{X}$ is an image in a batch $\mathcal{B}$ and $c \in \mathcal{C} = \{REAL, FS, FR, EFS, FE\}$ is the type of the image. Here, $d^f$ and $d^e$ represent text descriptions of the *facial* and *environment* attributes of $x$, respectively. $\mathcal{E}_I$ and $\mathcal{E}_T$ are the CLIP image and text encoders.

### 4.1   ENHANCED CLIP VISION EMBEDDINGS

**Cross-Layer Attention Module.**   Although previous studies leverage the final layer's output from the CLIP vision encoder for image representation, our investigation (Sec. 5.3) reveals that relying solely on the final representation is suboptimal. It struggles to capture low-level artifacts critical to distinguish fake images (Koutlis & Papadopoulos, 2024; Yan et al., 2025a). To address this, we introduce the Cross-Layer Attention Module (CLAM), which offers two significant advantages for visual representation: (i) it fuses multi-level features from all encoder layers, capturing both high-level semantics and low-level artifacts to construct a richer visual representation than using

the final layer alone; and (ii) it adopts a self-attention mechanism that dynamically highlights the most relevant lower-level features and guides them to achieve the objective earlier, benefiting the detector's generalization (Zhang et al., 2022). Specifically, we first extract the [CLS] token from each of the $L$ transformer layers within the CLIP image encoder $\mathcal{E}_I$, denoted as $f_{\text{cls}}^{(l)} \in \mathbb{R}^D$ for layer $l \in \{1, \ldots, L\}$. This collection of tokens, $f_{\text{cls}} = [f_{\text{cls}}^{(1)}, f_{\text{cls}}^{(2)}, \ldots, f_{\text{cls}}^{(L)}] \in \mathbb{R}^{L \times D}$, forms a sequence of hierarchical features. We then employ a multihead self-attention mechanism (Vaswani et al., 2017) over these features to capture the inter-dependencies across all transformer encoder blocks:

$$f_{\text{sat}}^{(h)} = \texttt{Attn}(f_{\text{cls}} W_Q^{(h)}, f_{\text{cls}} W_K^{(h)}, f_{\text{cls}} W_V^{(h)}), \quad f_{\text{sat}} = \text{Concat}([f_{\text{sat}}^{(h)}]_{h=1}^H) \cdot W_O, \tag{1}$$

where $W_Q^{(h)}, W_K^{(h)}, W_V^{(h)}$, and $W_O$ are learnable projection matrices, and $H$ is the number of attention heads. The resulting feature sequence $f_{\text{sat}}$ is first aggregated into a single vector, $f'_{\text{sat}}$, via global average pooling. This single vector $f'_{\text{sat}}$ is then concatenated with the model's final representation, $f_{\text{final}}$, to leverage both low-level and high-level information. The combined vector is further refined using a lightweight adapter, composed of two linear layers with a GELU activation (Hendrycks & Gimpel, 2016), to produce the final visual representation $f$ as follows:

$$f = W_2 \cdot \text{GELU}\big(W_1 \cdot \text{Concat}\left([f'_{\text{sat}}, f_{\text{final}}]\right)\big), \tag{2}$$

where $W_1$ and $W_2$ are learnable weights.

## 4.2 FINE-GRAINED DEEPFAKE CLASSIFICATION

**Image-Text Classification Loss.** During the training phase, we randomly sample the set of the text labels $\mathbf{T} = \{t_{REAL}, t_{FS}, t_{FR}, t_{EFS}, t_{FE}\}$ for each type once per batch. After extracting the text embedding $u_{c_i} = \mathcal{E}_T(t_{c_i})$ from text label $t_{c_i}$, we compute the cosine similarity between the image and the text embeddings. The similarity score serves as a logit and is passed to the cross-entropy loss as follows:

$$\mathcal{L}_{\text{C}} = -\sum_{i=1}^{|\mathcal{B}|} \log \left( \frac{\exp\left(\tau \cdot \text{sim}(f_i, u_{c_i})\right)}{\sum_{j=1}^{|\mathcal{C}|} \exp\left(\tau \cdot \text{sim}(f_i, u_{c_j})\right)} \right), \tag{3}$$

where $\text{sim}(a, b) = \frac{a \cdot b^T}{\|a\| \|b\|}$ and $\tau$ is a learnable logit scaling parameter initialized following CLIP (Radford et al., 2021).

**Text Label Separation Loss.** While humans can distinguish our manually curated text labels, they may not be sufficiently separated in the CLIP embedding space. We observe that the cosine similarities among different text label embeddings remain high, which can hinder classification. To address this, we introduce a separation loss that enforces greater distinctiveness among text label embeddings, thereby improving classification. We compute the cosine similarity between each pair of text label embeddings and enforce the similarity matrix to be closer to the identity matrix. This separation loss is formulated as:

$$\mathcal{L}_{\text{S}} = \|\text{sim}(u, u^T) - \mathbb{I}\|_F^2, \tag{4}$$

where $\mathbb{I}$ is the identity matrix and $\| \cdot \|_F$ the Frobenius norm.

## 4.3 DUAL IMAGE-TEXT CONTRASTIVE LEARNING

**Multi-Aspect Semantic Alignment.** We employ a dual contrastive loss to align each image with both its corresponding *facial* and *environmental* descriptions. For each image, we construct two distinct positive pairs: one with its facial description and the other with its environmental description. All other text descriptions within the batch serve as negatives for both alignment objectives. This dual supervision provides two key advantages: (i) it encourages the model to jointly attend to both detailed facial attributes and broader contextual elements within the scene; and (ii) it encourages disentanglement between forensic signals and non-forensic contextual elements, helping the model avoid spurious forgery cues.

**Efficient Contrastive Learning with SigLIP Loss.** For our image-text alignment, we adopt the SigLIP loss (Zhai et al., 2023) to avoid the substantial computational cost of conventional softmax-based losses (*e.g.,* InfoNCE), which demand massive batch sizes (32K in CLIP (Radford et al., 2021)). SigLIP utilizes a sigmoid-based binary cross-entropy loss and efficiently gathers negative samples by swapping text embeddings across GPUs, enabling strong performance without relying on large-scale batches. Given a text description $d_i^\kappa$, where $\kappa \in \{f, e\}$, we extract the corresponding text embedding $v_i^\kappa = \mathcal{E}_T(d_i^\kappa)$. The SigLIP loss between image and text embeddings is given by:

$$\mathcal{L}_\mathrm{D} = -\sum_{i=1}^{|\mathcal{B}|} \sum_{j=1}^{|\mathcal{B}_\mathrm{all}|} \left[ \log \sigma \bigg( z_{ij}(\tau \cdot \mathrm{sim}(f_i, v_j^f) + b) \bigg) + \log \sigma \bigg( z_{ij}(\tau \cdot \mathrm{sim}(f_i, v_j^e) + b) \bigg) \right], \quad (5)$$

where $f_i$ denotes the image embedding for the $i$-th sample in the local mini-batch $\mathcal{B}$, while $v_j^f$ and $v_j^e$ denote text embeddings in $\mathcal{B}_\mathrm{all}$, the global batch formed by gathering text embeddings from all GPUs. Specifically, $v_j^f$ and $v_j^e$ are derived from $d_j^f$ and $d_j^e$, respectively. The similarity between image and text embeddings is scaled by reusing the logit scaling parameter $\tau$ from the classification loss, along with a bias term $b$ initialized to $-\tau$. The binary indicator $z_{ij} \in \{-1, +1\}$ specifies whether the image-text pair $(f_i, v_j)$ is positive or negative, thereby determining the direction of the logit. The final logit is then passed through a sigmoid function $\sigma(\cdot)$ to compute the contrastive loss.

### 4.4 OVERALL OBJECTIVE FUNCTION

The overall loss combines the classification and regularization terms, promoting both accurate classification and meaningful image-text alignment. The loss is defined as:

$$\mathcal{L}_{\mathcal{SD}^2} = \alpha \cdot \mathcal{L}_\mathrm{C} + \mathcal{L}_\mathrm{S} + \mathcal{L}_\mathrm{D}, \quad (6)$$

where $\alpha$ is a hyperparameter controlling the contribution of the classification loss term. We present the pseudo-code for the $\mathcal{SD}^2$ implementation in Alg. 1 of the Appendix.

## 5 EXPERIMENTS

### 5.1 EXPERIMENTAL SETTINGS

**Training Details.** For training our large-scale multi-modal dataset, we adopt CLIP-ViT-L/14 (Ilharco et al., 2021) as the backbone for visual-language training. The model is optimized using AdamW (Loshchilov & Hutter, 2017) at a learning rate of 5e-7 and a batch size of 1,024 through gradient accumulation over 4 epochs. To efficiently fine-tune the model, we apply Low-Rank Adaptation (LoRA) (Hu et al., 2022) to both the image and text encoders, with hyperparameters set as follows: $r_\mathrm{lora} = 8$, $\alpha_\mathrm{lora} = 32$, and $p_\mathrm{dropout\text{-}lora} = 0.1$. We employ data augmentation techniques from CLIP training for deepfake detection (Yan et al., 2024), including random rotation, isotropic resizing, brightness and contrast adjustments, and compression. The overall objective function uses a hyperparameter $\alpha = 2.0$. Our method is implemented in PyTorch (Paszke et al., 2019) on 8 NVIDIA GeForce RTX 3090 GPUs.

**Testing Details.** During testing, we use the most straightforward text label from the predefined text label set for each of the five types (*e.g., Face Swapping*: `"a photo of a Face Swapping"`). The image feature is then matched with the most similar text feature based on cosine similarity to compute the classification logits. The final classification is based on the logits corresponding to each label. If the logit for real is the highest, the image is classified as real. Otherwise, it is classified as fake.

### 5.2 EXPERIMENTAL RESULTS

#### 5.2.1 FACIAL DEEPFAKE DETECTION

**Evaluation Protocols.** We adopt six baseline models: (1) Xception (Chollet, 2017), a CNN detecting subtle distortions with separable convolutions, (2) RECCE (Cao et al., 2022), a spatial-domain

method tracing forgery via reconstruction, (3) SPSL (Liu et al., 2021a), a frequency-domain approach finding artifacts with phase-based learning, (4) CLIP (Radford et al., 2021), a vision foundation model, fully fine-tuned with a linear head for classification, (5) CLIP-LoRA (Radford et al., 2021; Hu et al., 2022), a CLIP adaptation with our LoRA settings, and (6) CLIP-SVD (Effort) (Yan et al., 2025b), a SOTA method that adapts CLIP via Singular Value Decomposition (SVD) to preserve pre-trained knowledge while learning forgery-specific features.

We conduct two evaluations to assess facial deepfake detection performance. **(i) Intra-domain evaluation:** To ensure the fairness, all baselines are trained on our integrated dataset, **MMI-DD**, and then evaluated across seven datasets, including DF40 (Yan et al., 2024), DFF (Song et al., 2023), DFFD (Dang et al., 2019), DFDCP (Dolhansky et al., 2020), KoDF (Kwon et al., 2021), FF++ (c40) (Rossler et al., 2019), and TIMIT (Korshunov & Marcel, 2018). Specifically, we divide DF40 into six subsets: *FF*, *CDF*, *FS*, *FR*, *EFS*, and *FE*. Both *FF* and *CDF* (32 fake types each) are derived from FaceForensics++ (Rossler et al., 2019) and Celeb-DF (Li et al., 2020) domain, respectively. *FS*, *FR*, *EFS*, and *FE* are organized based on manipulation types, irrespective of source domains. *FS*, *FR*, *EFS*, and *FE* subsets include 9, 12, 10, and 1 manipulation types, respectively. This evaluation assesses model effectiveness within the training domain and measures how well each method leverages large-scale training data. **(ii) Cross-domain evaluation:** We use the same trained models and test them on challenging real-world benchmarks, including WildDeepFake (Zi et al., 2020), UADFV (Li et al., 2018), DFDC (Dolhansky et al., 2020), and DF40-Test subset from the DF40 (Yan et al., 2024). Among these DF40-Test subsets, *FS* and *FR* include fake data from a single manipulation type, while *EFS* and *FE* contain 2 and 4 types of fake data, respectively. A detailed description of DF40-Test is provided in Sec. A.2 of the Appendix. This experiment measures the generalization ability of each method to unseen distributions.

Table 2: Intra-domain detection performance (AUC). The best results are highlighted in **bold**.

| Methods | Detector Type | DF40 | | | | | | DFF | DFFD | DFDCP | KoDF | FF++ (c40) | mAUC |
|---|---|---|---|---|---|---|---|---|---|---|---|---|---|
| | | *FF* | *CDF* | *FS* | *FR* | *EFS* | *FE* | | | | | | |
| Xception | Naive | 97.15 | 93.39 | 91.87 | **95.66** | 96.29 | 99.28 | 91.25 | 96.73 | 97.43 | 96.53 | 78.63 | *94.02* |
| RECCE | Spatial | 95.23 | 92.08 | 90.81 | 94.18 | 96.14 | 99.69 | 89.01 | 94.93 | 98.21 | 92.28 | 78.27 | *92.80* |
| SPSL | Frequency | **97.59** | 93.98 | 93.45 | 95.62 | 97.26 | 99.63 | 92.50 | 95.70 | 98.71 | 95.59 | **80.40** | *94.58* |
| CLIP | CLIP-based | 61.72 | 57.28 | 55.61 | 67.58 | 54.50 | 65.39 | 62.32 | 73.55 | 79.61 | 59.05 | 52.84 | *62.68* |
| CLIP-LoRA | CLIP-based | 71.86 | 69.16 | 65.84 | 75.86 | 66.37 | 91.69 | 86.66 | 86.30 | 91.90 | 95.30 | 57.35 | *78.03* |
| CLIP-SVD (Effort) | CLIP-based | 88.54 | 87.53 | 84.30 | 86.50 | 92.57 | 95.53 | 84.33 | 97.74 | 90.95 | 81.84 | 72.52 | *87.48* |
| $\mathcal{SD}^2$ (*ours*) | CLIP-based | 97.52 | **96.66** | **97.61** | 94.23 | **98.94** | **99.73** | **98.02** | **99.88** | **99.66** | **97.08** | 74.09 | ***95.76*** |

Table 3: Cross-domain detection performance (AUC).

| Methods | Detector Type | UADFV | WildDeepFake | DFDC | DF40-Test | | | | mAUC |
|---|---|---|---|---|---|---|---|---|---|
| | | | | | *FS* | *FR* | *EFS* | *FE* | |
| Xception | Naive | 69.40 | 70.43 | 60.46 | 96.76 | 92.97 | 44.63 | 76.04 | *72.67* |
| RECCE | Spatial | 58.78 | 64.62 | 60.04 | 94.34 | 90.59 | 47.27 | 80.22 | *70.84* |
| SPSL | Frequency | 60.37 | 69.61 | 60.11 | 95.92 | **95.51** | 64.55 | 83.71 | *75.68* |
| CLIP | CLIP-based | 54.98 | 58.08 | 55.77 | 96.39 | 87.96 | 59.44 | 76.22 | *69.83* |
| CLIP-LoRA | CLIP-based | 66.55 | 69.41 | 69.44 | 92.74 | 84.33 | 40.75 | 81.62 | *72.12* |
| CLIP-SVD (Effort) | CLIP-based | 95.47 | 75.89 | 69.57 | 97.07 | 78.88 | 60.60 | 96.41 | *81.98* |
| $\mathcal{SD}^2$ (*ours*) | CLIP-based | **96.44** | **82.80** | **80.36** | **99.29** | 84.49 | **74.06** | **97.13** | ***87.79*** |

**Results.** Experimental results reveal two key strengths of our approach. As shown in Tab. 2, $\mathcal{SD}^2$ achieves superior overall intra-domain detection, with an mAUC of 95.76%, outperforming all baselines in the majority of the 11 subsets. Additionally, Tab. 3 highlights robust cross-domain generalization, with an mAUC of 87.79%, surpassing baselines on challenging unseen datasets. These results confirm $\mathcal{SD}^2$'s capacity to generalize without overfitting, excelling in both settings.

### 5.2.2 GENERAL SYNTHETIC IMAGE DETECTION

Although our model is primarily trained on facial image-text pairs, **we explore its ability to detect synthetic content beyond facial manipulations**.

**Evaluation Protocols.** We utilize only the vision encoder of our model and perform linear probing for binary classification. All other components are frozen, and only the linear layer is trained. This setup follows the same approach as UnivFD (Ojha et al., 2023), which uses linear probing with frozen CLIP for synthetic image detection. To ensure a fair comparison, our model and all baselines follow the same protocol (Zhu et al., 2023b), fine-tuning on SD v1.4 fake images and ImageNet real images. Generalization performance is then evaluated on test sets containing 8 generative models. We measure performance using accuracy (ACC), consistent with (Yan et al., 2025a; Zhu et al., 2023a), with a classification threshold of 0.5 for fair comparison.

Table 4: Cross-model evaluation performance (ACC) on the GenImage dataset. While the results are directly sourced from (Yan et al., 2025a), we additionally implement CLIP-SVD, *a.k.a.* Effort, from (Yan et al., 2025b) following its official code.

| Methods | Midjourney | SD v1.4 | SD v1.5 | ADM | GLIDE | Wukong | VQDM | BigGAN | mACC |
|---|---|---|---|---|---|---|---|---|---|
| ResNet-50 (He et al., 2016) | 54.90 | **99.90** | 99.70 | 53.50 | 61.90 | 98.20 | 56.60 | 52.00 | *72.09* |
| DeiT-S (Touvron et al., 2021) | 55.60 | **99.90** | 99.80 | 49.80 | 58.10 | 98.90 | 56.90 | 53.50 | *71.56* |
| Swin-T (Liu et al., 2021b) | 62.10 | **99.90** | 99.80 | 49.80 | 67.60 | 99.10 | 62.30 | 57.60 | *74.78* |
| CNNSpot (Wang et al., 2020) | 52.80 | 96.30 | 95.90 | 50.10 | 39.80 | 78.60 | 53.40 | 46.80 | *64.21* |
| Spec (Zhang et al., 2019) | 52.00 | 99.40 | 99.20 | 49.70 | 49.80 | 94.80 | 55.60 | 49.80 | *68.79* |
| F3Net (Qian et al., 2020) | 50.10 | **99.90** | **99.90** | 49.90 | 50.00 | **99.90** | 49.90 | 49.90 | *68.69* |
| GramNet (Liu et al., 2020) | 54.20 | 99.20 | 99.10 | 50.30 | 54.60 | 98.90 | 50.80 | 51.70 | *69.85* |
| DIRE (Wang et al., 2023a) | 60.20 | **99.90** | 99.80 | 50.90 | 55.00 | 99.20 | 50.10 | 50.20 | *70.66* |
| UnivFD (Ojha et al., 2023) | 73.20 | 84.20 | 84.00 | 55.20 | 76.90 | 75.60 | 56.90 | **80.30** | *73.29* |
| GenDet (Zhu et al., 2023a) | **89.60** | 96.10 | 96.10 | 58.00 | 78.40 | 92.80 | 66.50 | 75.00 | *81.56* |
| PatchCraft (Zhong et al., 2023) | 79.00 | 89.50 | 89.30 | 77.30 | 78.40 | 89.30 | 83.70 | 72.40 | *82.30* |
| AIDE (Yan et al., 2025a) | 79.38 | 99.74 | 99.76 | **78.54** | **91.82** | 98.65 | 80.26 | 66.89 | *86.88* |
| CLIP-SVD (Effort) (Yan et al., 2025b) | 71.70 | 99.90 | 99.71 | 63.35 | 66.64 | 99.20 | 86.19 | 52.84 | *79.94* |
| $\mathcal{SD}^2$ (***ours***) | 85.76 | 98.45 | 98.23 | 68.25 | 88.59 | 98.27 | **91.15** | 79.34 | ***88.50*** |

**Results.** Results on the GenImage dataset are presented in Tab. 4. Our method achieves mACC of 88.50% across 8 test sets. Compared to other CLIP-based baselines UnivFD and CLIP-SVD (Effort), our approach surpasses them by 15.21% and 8.56%, respectively. Furthermore, when compared to the SOTA method AIDE, our method outperforms AIDE by 1.62%. This highlights the superior generalization capabilities of $\mathcal{SD}^2$ for synthetic image detection.

## 5.3 ABLATION STUDY

**Classification Loss.** The classification loss, comprising image-text classification and text label separation objectives (Sec. 4.2), is critical for training convergence. Without enforcing orthogonality among the text label embeddings, training fails to stabilize. Thus, we jointly term these as classification loss here. As shown in the first row of Tab. 5, applying our classification loss yields 80.24% AUC, significantly enhancing performance. Comparisons to cross-domain evaluation results in Tab. 3, our method outperforms baselines such as Xception, RECCE, and SPSL. Notably, it also improves performance over CLIP and CLIP-LoRA, even though they share the same CLIP-ViT-L/14 backbone. This result suggests that pre-trained backbones require tailored objectives for deepfake detection.

**Leveraging Intermediate Features.** We further assess the impact of incorporating intermediate features from the CLIP image encoder through our CLAM (Sec. 4.1). The second row of Tab. 5 shows a 1.6% AUC increase, emphasizing the benefit of leveraging multi-level features. Additionally, we provide a qualitative comparison regarding the presence of CLAM in Appendix Sec. C.

**Contrastive Loss.** Our dual image-text contrastive loss with VLM-generated descriptions (Sec. 4.3), significantly boosts detection performance. As detailed in Tab. 5 (rows 3–4), jointly leveraging two contrastive losses: $\mathcal{L}_\mathrm{D}$-*Face* using facial descriptions and $\mathcal{L}_\mathrm{D}$-*Env* using environmental descriptions achieves the highest performance gain of 5.95%. While individual semantic cues are beneficial, their joint application proves optimal for robust generalization.

**Combining All Components.** Combining all components yields the highest AUC of 87.79%, as shown in the final row of Tab. 5. Each component contributes incrementally, with their synergy maximizing performance. Thus, we adopt this configuration as our final model.

Table 5: Ablation studies analyzing the impact of key components. All models are trained on our integrated dataset and tested on four cross-domain datasets in Sec. 5.2.1.

| $\mathcal{L}_C$ | CLAM | $\mathcal{L}_D$-*Face* | $\mathcal{L}_D$-*Env* | mAUC |
|---|---|---|---|---|
| ✓ | - | - | - | *80.24* |
| ✓ | ✓ | - | - | *81.84* |
| ✓ | ✓ | ✓ | - | *84.34* |
| ✓ | ✓ | ✓ | ✓ | **87.79** |

Table 6: Performance (mAUC) across training data scales, from *100K* to *3M* (full dataset).

| Methods | Scale of Training Data | | |
|---|---|---|---|
| | *100K* | *1M* | *3M* |
| Xception | 71.11 | 71.69 | 72.96 |
| RECCE | 69.71 | 69.28 | 70.84 |
| SPSL | 71.56 | 73.84 | 75.68 |
| CLIP | 69.82 | 69.74 | 69.83 |
| CLIP-LoRA | 78.84 | 78.40 | 72.12 |
| CLIP-SVD (Effort) | 82.27 | 82.00 | 81.98 |
| $\mathcal{SD}^2$ (***ours***) | **82.90** | **84.85** | **87.79** |

**Scale of Training Dataset.** We train our proposed $\mathcal{SD}^2$ and baselines with 100K, 1M, and 3M images from our integrated dataset, assessing cross-domain performances as in Sec. 5.2.1 with mAUC reported in Tab. 6. Our findings reveal that **dataset scale alone does not guarantee performance gains**. Conventional methods such as Xception, RECCE, and SPSL exhibit early saturation. Similarly, strong vision foundation models like CLIP, CLIP-LoRA, and CLIP-SVD (Effort) also struggle to scale, with performance plateauing or even degrading as data increases. In contrast, $\mathcal{SD}^2$ not only outperforms all baselines but also continues to improve with more data. This suggests our multimodal strategy is key to leveraging large-scale datasets, addressing a limitation of prior approaches.

**Hyperparameter Sensitivity.** We conduct additional experiments on hyperparameters to assess the robustness of $\mathcal{SD}^2$. All experiments in this section are trained on a 1M-sized subset of the MMI-DD dataset, with performance measured by cross-domain mAUC, following the protocol described in Sec. 5.2.1. We first analyze the effect of the weighting factor $\alpha$ introduced in Eq. 6. As shown in Tab. 7, $\mathcal{SD}^2$ achieves consistently strong performance across different $\alpha$ values, indicating low sensitivity to this factor. We further evaluate the impact of the LoRA rank by varying $r_{\text{lora}} \in \{4, 8, 16\}$ while keeping the other LoRA hyperparameters fixed ($\alpha_{\text{lora}} = 32$, $p_{\text{dropout-lora}} = 0.1$). The results demonstrate stable performance across rank variations, supporting the architectural robustness of the proposed framework. Based on these results, we adopt the best-performing configuration in the final model.

Table 7: Hyperparameter sensitivity on cross-domain mAUC.

| Hyperparameter | mAUC |
|---|---|
| $\alpha = 1.5$ | *83.45* |
| $\alpha = 2.0$ | ***84.85*** |
| $\alpha = 2.5$ | *83.02* |
| $r_{\text{lora}} = 4$ | *82.97* |
| $r_{\text{lora}} = 8$ | ***84.85*** |
| $r_{\text{lora}} = 16$ | *84.37* |

## 6    CONCLUSION

In many cases, existing deepfake detectors remain tied to single-dataset training, limiting their ability to leverage diverse available data resources and build comprehensive knowledge. Additionally, as training data scales up, they struggle to consolidate heterogeneous distributions, leading to a performance drop. To address these challenges, we unify 3.6M facial images from 11 datasets, namely MMI-DD, enriched with fine-grained type annotation and VLM-generated descriptions. Building on this, we propose $\mathcal{SD}^2$, a scalable vision-language framework for deepfake detection. $\mathcal{SD}^2$ enhances CLIP with the Cross-Layer Attention Module (CLAM) to capture multi-level visual features, and is optimized with three complementary objectives: type-aware classification, type embedding separation, and semantic alignment. Extensive experiments show that $\mathcal{SD}^2$ achieves SOTA performance across facial and non-facial domains, while exhibiting strong scalability as data volume increases.

## ACKNOWLEDGEMENT

This work was partly supported by Institute for Information & communication Technology Planning & evaluation (IITP) grants funded by the Korean government MSIT: (RS-2022-II220688, RS-2019-II190421). Also, this work was supported by the Cyber Investigation Support Technology Development Program (No.RS-2025-02304983) of the Korea Institute of Police Technology (KIPoT),

funded by the Korean National Police Agency. Lastly, this work was supported by the National Research Foundation of Korea (NRF) grant funded by the Korea government (MSIT) (No. RS-2024-00356293).

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

APPENDIX

## GENAI USAGE DISCLOSURE

We employed GPT-5 for text rephrasing, grammar checking, and code debugging. The authors reviewed and modified all outputs. All ideas and content were originally written and verified by the authors.

## A  DATASETS

Here, we describe the various benchmark deepfake datasets that we have integrated. These datasets are utilized to train and evaluate our proposed model. We also detail the preprocessing steps applied to the collected datasets.

### A.1  TRAINING DATASETS

- **DF40.** DF40 (Yan et al., 2024) is a comprehensive deepfake dataset proposed by Yan et al., incorporating a variety of the latest forgery techniques. It features 40 distinct deepfake generation and editing methods, making it a valuable resource for evaluating detection performance. The dataset is created from popular real image datasets, including FaceForensics++ (Rossler et al., 2019) and Celeb-DF (Li et al., 2020), ensuring diverse and representative content. These deepfake samples are categorized into four manipulation types: Face Swapping (FS), Face Reenactment (FR), Entire Face Synthesis (EFS), and Face Editing (FE). The DF40 dataset consists of a total of 77,200 real images and 1,613,000 fake images, providing a robust and extensive collection for evaluation.

- **DFF.** DeepFakeFace (DFF) (Song et al., 2023) consists of 30,000 real images and 90,000 fake images. The real images are collected by sorting celebrity face photos extracted from the IMDB-WIKI (Rothe et al., 2018) dataset using RetinaFace (Deng et al., 2019), ensuring high-quality and diverse real samples. The fake images are generated using three different techniques: Stable Diffusion v1.5 (Rombach et al., 2022), Stable Diffusion Inpainting (Rombach et al., 2022), and InsightFace (Ren et al., 2023), representing a variety of deepfake generation methods.

- **DFFD.** Diverse Fake Face Dataset (DFFD) (Dang et al., 2019) consists of 58,703 real images, 240,336 fake images, 1,000 real video clips, and 3,000 fake video clips. The real images in DFFD are sourced from the FFHQ (Karras et al., 2019) and CelebA (Liu et al., 2015) datasets, while both real and fake videos are derived from the FaceForensics++ (Rossler et al., 2019) dataset. The dataset includes fake videos generated using four distinct categories of methods: identity swap, expression swap, attribute manipulation, and entire face synthesis.

- **DFDCP.** DFDCP (Dolhansky et al., 2020) is a preview version of the DeepFakes Detection Challenge (DFDC) dataset, consisting of approximately 5,000 videos, including both original and manipulated content. The dataset is constructed with careful consideration of diversity across various aspects such as gender, skin color, and age. The original videos encompass a wide range of lighting environments and head poses.

- **KoDF.** Korean DeepFake Detection Dataset (KoDF) (Kwon et al., 2021) consists of 62,166 real video clips of 90 seconds and 175,776 fake clips of 15 seconds or longer. The fake clips in the KoDF dataset are generated using six distinct forgery methods: FaceSwap (FaceSwap, 2016), DeepFaceLab (DeepFaceLab, 2023), FSGAN (Nirkin et al., 2019), FOMM (Siarohin et al., 2019), ATFHP (Yi et al., 2020), and Wav2Lip (Prajwal et al., 2020). Specifically, FaceSwap, DeepFaceLab, and FSGAN are face-swapping models, while FOMM is a video-driven face-reenactment model. ATFHP and Wav2Lip are audio-driven face-reenactment models.

- **FF++ (c40).** FaceForensics++ (FF++) (Rossler et al., 2019) is a widely used face forgery detection dataset consisting of 1,000 real videos and 4,000 fake videos. The fake videos are generated using both graphics-based manipulation techniques (Face2Face (Thies et al., 2016), FaceSwap (FaceSwap, 2016)) and learning-based methods (DeepFakes (DeepFakes,

2017), FaceShifter (Li et al., 2019), and NeuralTextures (Thies et al., 2019)). FF++ is provided in three versions based on the compression level: raw, lightly compressed (c23), and heavily compressed (c40). Since the aforementioned DF40 dataset includes the c23 version, we add the c40 version for our experiments.

- **TIMIT.** DeepfakeTIMIT (TIMIT) (Korshunov & Marcel, 2018) consists of fake videos generated from the real video sequences of the VidTIMIT (Sanderson & Lovell, 2009) dataset using a GAN-based face-swapping method. The authors of the TIMIT dataset trained two types of fake video generation models: one using a GAN model to generate low-quality fake videos (LQ) and another using a GAN model to generate high-quality fake videos (HQ). The dataset contains a total of 640 fake videos, with 320 videos of low quality and 320 videos of high quality.

- **CelebA.** CelebA (Liu et al., 2015) is a large-scale face attributes dataset containing 202,599 images of 10,177 celebrities. Each image is meticulously labeled by a professional team, providing 40 binary attributes and 5 facial landmarks (both eyes, nose, and the corners of the mouth).

- **CelebA-HQ.** CelebA-HQ (Karras et al., 2017) is a high-quality version of the widely used CelebA dataset. The CelebA-HQ dataset consists of 30,000 images selected from the original CelebA dataset, which have been enhanced to a resolution of $1,024 \times 1,024$. The image enhancement process begins with enlarging the original CelebA images to $4,096 \times 4,096$ using the mirror padding method, followed by Gaussian filtering to smooth the image. The enlarged image is then cropped to a size of $1,024 \times 1,024$, centered on the face, using landmark positions from the CelebA dataset. This process is applied to a total of 202,599 images from the original CelebA dataset, with the top 30,000 images selected based on a frequency-based quality metric to form the CelebA-HQ dataset.

- **CelebV-HQ.** CelebV-HQ (Zhu et al., 2022) consists of 35,666 video clips, totaling approximately 65 hours of footage, sourced from 15,653 unique individuals. All videos maintain a minimum resolution of $512 \times 512$ and have durations ranging from 3 to 20 seconds. The dataset is curated with 83 manually labeled facial attributes, covering aspects of appearance, behavior, and emotion, ensuring a diverse and well-annotated collection.

- **FFHQ.** FFHQ (Karras et al., 2019) consists of 70,000 high-resolution images of $1024 \times 1024$, providing a diverse set of human faces with variations in age, ethnicity, and accessories such as glasses and hats. The images are sourced from `https://www.flickr.com/` and are automatically aligned and cropped using dlib (King, 2009).

## A.2 EVALUATION DATASETS

- **WildDeepFake.** WildDeepFake (Zi et al., 2020) is a dataset designed to address the limitations of existing deepfake detection models in real-world scenarios. It consists of 7,314 face images extracted from 707 fake videos, providing a diverse and challenging benchmark for evaluating detection performance. Unlike prior datasets with limited diversity and low image quality, WildDeepFake captures a wide range of deepfake variations by collecting videos from the Internet.

- **UADFV.** UADFV (Li et al., 2018) consists a total of 96 videos, including 49 real and 49 fake videos, providing a balanced set for evaluating detection models. The average duration of videos in the dataset is approximately 11.14 seconds, ensuring a consistent temporal structure across samples.

- **DFDC.** DFDC (Dolhansky et al., 2020) is introduced as part of the DeepFake Detection Challenge (DFDC), consisting of 128,154 video clips. It encompasses a diverse range of deepfake generation techniques along with refinement methods to enhance the realism of manipulated content. To further increase the complexity and robustness of the data set, the validation and test sets incorporate additional augmentations such as distractor, which overlays various objects, and augmenter, which applies geometric and color transformations as well as frame rate modifications.

- **DF40-Test.** DF40-Test (Yan et al., 2024) is a specialized evaluation dataset created by aggregating the test-only subsets defined in DF40 (Yan et al., 2024). To ensure a comprehensive assessment of deepfake detection models, the original authors curated distinct test

sets for each manipulation type. We combine these test-only datasets into a unified version, which is utilized for our evaluation. This dataset includes fake images generated using various state-of-the-art methods, such as DeepFaceLab (DeepFaceLab, 2023) (for *FS* type), HeyGen (HeyGen., 2020) (for *FR* type), MidJourney6 (MidJourney., 2022) and Whichis-Real (WhichFaceisReal., 2019) (for *EFS* type), and CollabDiff (Huang et al., 2023), Star-GAN (Choi et al., 2018), StarGANv2 (Choi et al., 2020), and StyleCLIP (Patashnik et al., 2021) (for *FE* type). The real images are sourced from UADFV (Li et al., 2018), VFHQ (Xie et al., 2022), and FFHQ (Karras et al., 2019), consisting of a total of 17,065 real images and 14,164 fake images.

## A.3 PREPROCESSING

We apply the following preprocessing steps to the collected datasets. First, we detect faces in the images (frames) using a Multi-task Cascaded Convolutional Networks (MTCNN) (Zhang et al., 2016). To reduce noise and ensure reliable face detection, we select only bounding boxes with a confidence score greater than 0.9. We then crop the images into square regions centered on the detected bounding boxes. For image datasets, all cropped images are used for training and evaluation. For video datasets, we sample 32 frames evenly spaced from the entire video. When the original dataset is already split into training, validation, and test sets, we use the training and validation sets for training. In cases where the dataset is not pre-split, we randomly shuffle the images (or video frames) in a 9:1 ratio and then sample them to form the training and test sets.

## A.4 DATA PARTITIONING PROTOCOLS

Integrating multiple deepfake datasets introduces the challenge of identity overlap due to shared original data sources, as detailed in Tab. 8. To prevent data leakage, we implement specific data partitioning protocols tailored to the nature of the overlap.

Table 8: Identity counts and original data sources for each dataset.

| Index | Dataset Name | Identity Count (Subjects) | Dataset Source |
|-------|--------------|---------------------------|----------------|
| 1 | DF40 | 1,059 | FaceForensics++, CDF |
| 2 | DFF | Unknown | IMDb-WIKI |
| 3 | DFFD | Unknown | FFHQ, CelebA, FaceForensics++ |
| 4 | DFDCP | 66 | Paid Actors |
| 5 | KoDF | 403 | Paid Korean Subjects |
| 6 | FF++ (c40) | 1,000 | YouTube Videos |
| 7 | TIMIT (LQ) | 32 | VidTIMIT |
| 8 | TIMIT (HQ) | 32 | VidTIMIT |
| 9 | CelebA | 10,177 | Web Crawled |
| 10 | CelebA-HQ | Unknown | CelebA |
| 11 | FFHQ | 70,000 | Flickr |

**Protocol 1: Managing Overlaps with Quality Variations (same source, different quality).** Several datasets, including DF40 & FF++ (c40), TIMIT (LQ) & TIMIT (HQ), and CelebA & CelebA-HQ, share identical source subjects but differ in visual quality (*e.g.,* compression level, resolution). For these cases, we utilize these variations as a form of natural data augmentation. Exposing the model to diverse quality levels for the same identity enhances robustness against compression artifacts and resolution degradation. To prevent data leakage, we follow the official training and testing splits provided by the original datasets (*e.g.,* FF++, TIMIT). For datasets without predefined splits (*e.g.,* CelebA, CelebA-HQ), we partition the data based on unique identity IDs, ensuring that no subject appears in both the training and testing sets.

**Protocol 2: Managing Overlaps within DFFD (same source, same quality).** The DFFD dataset is constructed from FFHQ, CelebA, and FaceForensics++, meaning its images share both source identities and visual quality with other datasets in our corpus. To prevent data duplicates, we remove

the FaceForensics++ subset. For the remaining subsets (originating from FFHQ and CelebA), we apply a random pre-compression process (JPEG quality 60–100) before storage. This strategy is designed to simulate the relationship between high-quality and low-quality forensic data, mimicking the domain shift observed between DF40 and FF++.

# B ACCURACY RESULTS FOR DEEPFAKE DETECTION

In Sec. 5.2.1 of our main paper, we report deepfake detection performance using Area Under the Curve (AUC). Here, we additionally present accuracy (ACC) results for a comprehensive analysis. For a fair comparison, we apply a classification threshold of 0.5 for all datasets. As shown in Tab. 9 and 10, our model exhibits superior overall performance (mACC) in both intra- and cross-domain evaluations, demonstrating its robust learning capacity on diverse training sets and strong generalization.

Table 9: Intra-domain detection performance (ACC).

| Methods | Detector Type | DF40 | | | | | | DFF | DFFD | DFDCP | KoDF | FF++ (c40) | TIMIT (HQ) | TIMIT (LQ) | mACC |
|---|---|---|---|---|---|---|---|---|---|---|---|---|---|---|---|
| | | FF | CDF | FS | FR | EFS | FE | | | | | | | | |
| Xception | Naive | 97.16 | 97.15 | 92.01 | **94.95** | 93.81 | **75.06** | 82.37 | 93.17 | 89.77 | 87.28 | 84.04 | **100.0** | **100.0** | 91.29 |
| RECCE | Spatial | 97.02 | 97.04 | 91.37 | 93.86 | 92.70 | 69.14 | 79.52 | 88.78 | 92.96 | 85.00 | 82.81 | **100.0** | **100.0** | 90.02 |
| SPSL | Frequency | 97.32 | 97.13 | 91.23 | 93.43 | 91.88 | 64.37 | 85.55 | 89.62 | 94.93 | 88.59 | **84.35** | **100.0** | **100.0** | 90.65 |
| CLIP | CLIP-based | 94.80 | 96.07 | 88.06 | 91.62 | 87.93 | 56.22 | 37.73 | 84.61 | 54.52 | 52.96 | 82.43 | 97.11 | 97.89 | 78.61 |
| CLIP-LoRA | CLIP-based | 96.12 | 96.85 | 89.09 | 91.94 | 89.93 | 56.89 | 80.22 | 88.72 | 84.68 | 89.80 | 83.14 | **100.0** | **100.0** | 88.26 |
| CLIP-SVD (Effort) | CLIP-based | 94.85 | 95.91 | 89.01 | 91.34 | 90.84 | 64.36 | 66.99 | 93.85 | 80.18 | 71.37 | 82.28 | 99.06 | 100.0 | 86.77 |
| $\mathcal{SD}^2$ (ours) | CLIP-based | **97.51** | **97.51** | **92.89** | 94.42 | **93.91** | 73.20 | **90.69** | **98.21** | **96.39** | **90.53** | 82.89 | **100.0** | **100.0** | **92.93** |

Table 10: Cross-domain detection performance (ACC).

| Methods | Detector Type | UADFV | WildDeepFake | DFDC | DF40-Test | | | | mACC |
|---|---|---|---|---|---|---|---|---|---|
| | | | | | FS | FR | EFS | FE | |
| Xception | Naive | 52.58 | 65.14 | 52.01 | 90.78 | **89.42** | 82.48 | 66.88 | 71.33 |
| RECCE | Spatial | 53.76 | 61.00 | 51.96 | 84.15 | 82.75 | 75.62 | 72.20 | 68.78 |
| SPSL | Frequency | 54.66 | 63.15 | 54.98 | 87.20 | 85.99 | 78.92 | 76.14 | 71.58 |
| CLIP | CLIP-based | 53.38 | 59.35 | 51.71 | 86.24 | 83.71 | 77.64 | 69.80 | 68.83 |
| CLIP-LoRA | CLIP-based | 55.90 | 55.31 | 60.84 | 76.36 | 73.79 | 66.42 | 74.59 | 66.17 |
| CLIP-SVD (Effort) | CLIP-based | **90.98** | 66.05 | 63.61 | 88.16 | 83.21 | 81.24 | 87.98 | 80.18 |
| $\mathcal{SD}^2$ (ours) | CLIP-based | 89.62 | **69.16** | **73.66** | **91.95** | 85.67 | **84.25** | **90.43** | **83.53** |

# C MORE ABLATION STUDIES

## C.1 TEXT LABEL SEPARATION LOSS

To analyze the impact of the proposed text label separation loss ($\mathcal{L}_S$) in Sec. 4.2, we conduct an ablation study on text label embeddings across the five defined types. We first compute the mean embeddings of text labels for each type and measure the cosine similarity between types. As shown in Fig. 4 (a), using the original CLIP text encoder results in high inter-type similarity (depicted in darker blue), indicating poor separability among manipulation types. In contrast, Fig. 4 (b) demonstrates that after applying the separation loss, our optimized model produces text embeddings that are nearly orthogonal across different types (with similarity approaching 0), leading to well-separated representations. These results highlight the effectiveness of our loss function in enforcing a more structured and discriminative embedding space.

Moreover, we measure the intra-type cosine similarity by randomly sampling five text labels from each type. Fig. 5 shows that intra-type cosine similarity is significantly higher (depicted in darker blue) when using our model's text encoder (b) compared to the original CLIP text encoder (a). This demonstrates that augmenting each type with semantically similar text labels enhances the model's semantic understanding of each manipulation type.

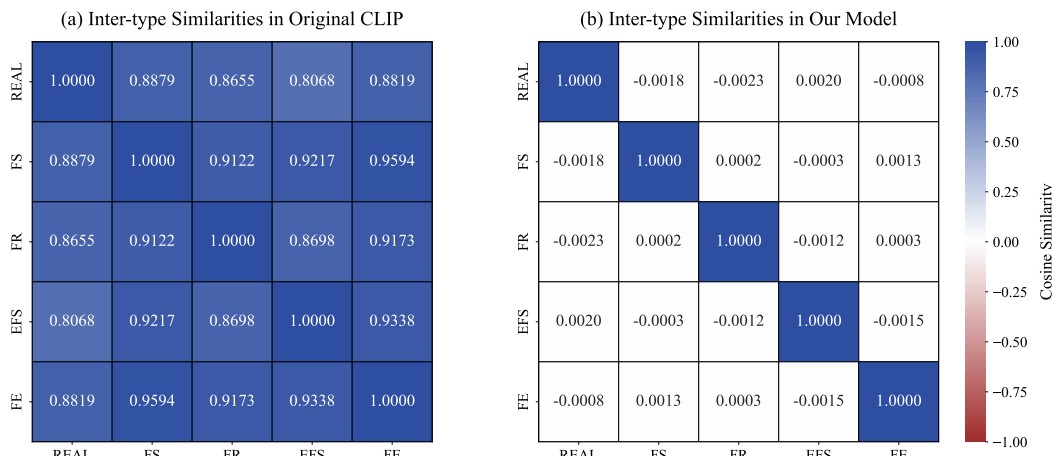

Figure 4: Cosine similarity between text labels across different types. (a) Using the original CLIP, the text labels across different types exhibit high similarity in the embedding space, preventing effective convergence of the intended type-guided separation. (b) With the application of our separation loss, we successfully enforce separation between different types, thereby optimizing the classification objective.

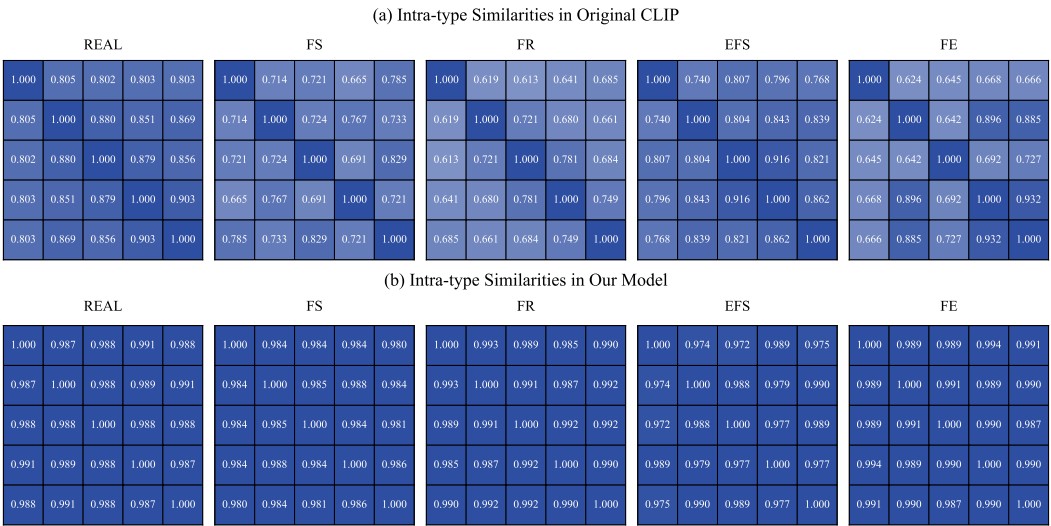

Figure 5: Cosine similarity between text labels within the same type. (a) In the original CLIP embedding space, our carefully crafted text labels exhibit low similarity with other text labels of the same type. (b) After optimization with our separation loss, the similarity between text labels of the same type increases significantly, demonstrating that the model has learned to capture the semantics of each type effectively.

## C.2 QUALITATIVE ANALYSIS OF CLAM

To qualitatively verify the impact of CLAM on feature representation, we visualized the image embeddings using t-SNE (Maaten & Hinton, 2008). Specifically, we randomly sampled 500 images from the DF40-Test (Yan et al., 2024) for this visualization. Figure 6 compares the feature spaces of our framework with and without the CLAM module. The results demonstrate that CLAM leads to significantly more separable and structured feature representations, effectively disentangling real and fake samples.

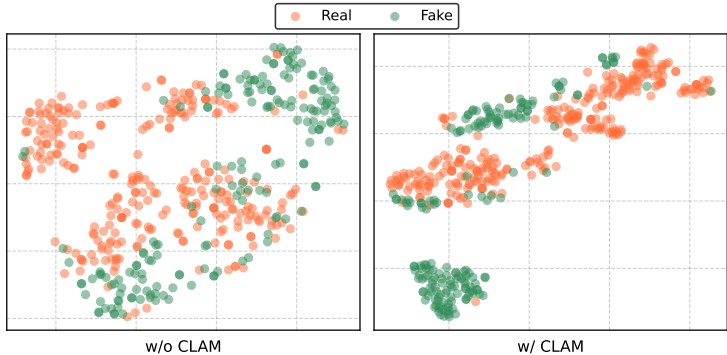

Figure 6: Impact of CLAM on feature representation. (Left) Without CLAM, real and fake embeddings are heavily entangled. (Right) With CLAM, the feature space exhibits enhanced separability, forming a more discriminative manifold with distinct clusters.

## D    PSEUDOCODE

We present the pseudocode implementation of our $\mathcal{SD}^2$ objective in Alg. 1.

---
**Algorithm 1** $\mathcal{SD}^2$ Pseudocode
---

```
# E_I: image encoder - Vision Transformer
# E_T: text encoder - Vision Transformer
# CLAM: Cross-Layer Attention Module
# GAP: Global Average Pooling
# adapter: learnable adapter for image encoder
# t_prime, b: temperature & bias of SigLIP
# x[B, C, H, W]: minibatch of images
# y[B]: minibatch of categorical labels
# c[B, L]: minibatch of augmented text labels
# d_f[B, L]: minibatch of facial descriptions
# d_e[B, L]: minibatch of environmental descriptions

# extract visual feature representations of images
f_final = E_I(x)
f_sat = CLAM(x)
f_sat_prime = GAP(f_sat)
f = adapter([f_final, f_sat_prime])

# extract all textual representations
u_c = E_T(c)
v_f, v_e = E_T(d_f), E_T(d_e)

# classification loss
logits = dot(f, u_c.T)
loss_c = cross_entropy_loss(logits, y)

# text label separation loss
u_n = l2_normalize(u_c)
loss_s = mean(power(dot(u_n, u_n.T)-np.eye(B), 2))

# dual contrastive loss
loss_d = siglip(f, v_f) + siglip(f, v_e)

def siglip(img, txt):
    t = exp(t_prime)
    z_i = l2_normalize(img)
    z_t = l2_normalize(txt)
    logits = dot(z_i, z_t.T) * t + b
    labels = 2 * eye(B) - ones(B)
    return -sum(log_sigmoid(labels * logits)) / B
```

---

## E    TEXT LABELS CONSTRUCTION

We manually curate text labels for five types: *REAL*, *Face Swapping*, *Face Reenactment*, *Entire Face Synthesis*, and *Face Editing*. Specifically, we input their definitions, descriptions, and a simple

prompt, `"a photo of a {type}"` into GPT-o1 (OpenAI, 2024) to generate 64 text labels per type. However, these generated text labels may misrepresent the type, lack sufficient coverage, or overlap ambiguously with other types. To refine them, we embed the text labels using the pre-trained CLIP text encoder (Radford et al., 2021) and compute the mean vector for each type. We then filter out half of the labels farthest from the mean vector. Subsequently, researchers manually inspect the remaining text labels, removing those with inadequate descriptions or redundancy across types. The final text labels used in training are listed in Tabs. 11–15.

## F  TEXT DESCRIPTION DETAILS

In this section, we describe the process of generating text descriptions for deepfake datasets, including detailed information on facial attributes and environmental factors. We also detail the design of the prompts used to generate these descriptions and describe the generation process. Finally, we present examples of the text descriptions from our datasets.

### F.1  PROMPT DESIGN FOR TEXT DESCRIPTIONS

Recent work (Wang et al., 2024a) leverages text descriptions tailored to Face Anti-Spoofing (FAS). Instead of relying on coarse-grained prompts, the authors improve generalization by aligning the visual and language components. Inspired by this approach, we generate text descriptions based on two key factors: facial attributes (age, gender, expression, and facial features) and environmental aspects (background and lighting). We directly adopt the facial and environmental prompts used in the referenced work to extract these features. The following prompts are used in our experiments:

- **Facial Description Prompt**:
  ```
  "Please describe the face (including age, gender,
  expression, appearance, etc.)  of the person in one
  sentence."
  ```

- **Environmental Description Prompt**:
  ```
  "Please describe the environment (including background,
  lighting intensity, etc.)  in one sentence."
  ```

### F.2  TEXT DESCRIPTION GENERATION VIA VLM

For generating text descriptions, we utilize InternVL2.5 (Chen et al., 2024), an open-source vision-language model with 1 billion parameters. We perform inference using the two aforementioned prompts for all images used in joint training and store the resulting responses. In line with findings from previous work (Zhang et al., 2024), we recognize that CLIP's effective token length often falls below 20 tokens, even though it is limited to 77 tokens. To mitigate this, we append the phrase `"in one sentence"` to each prompt and set the maximum length of the textual token sequence to 60. Our VQA inference using InternVL is implemented in PyTorch and performed on 4 NVIDIA GeForce RTX 3090 GPUs. Examples of our annotated data are visualized in Fig. 7.

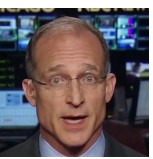

**Type:** REAL

**Face:** *The individual appears to be middle-aged, male, wearing glasses, and has a professional appearance in a well-lit news broadcast studio with a background of monitors.*

**Environment:** *The environment has a busy television studio background with various lights and screens, likely from a news broadcast. The lighting appears to be well-lit, illuminating the man clearly for his report.*

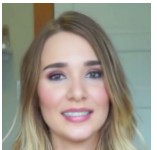

**Type:** Face Swapping

**Face:** *The woman in the portrait has fair complexion, light skin, shoulder-length wavy hair, light pink blush on her cheeks, full lips, and a friendly, radiant smile.*

**Environment:** *In the photo, the background is an indoor setting with light-colored walls and a door visible, and natural light is illuminating the scene.*

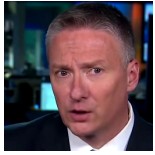

**Type:** Face Reenactment

**Face:** *The person appears to be middle-aged with short, graying hair. They have a serious or neutral expression, wearing a suit.*

**Environment:** *The man is in a newsroom with a blurred background. The lighting is typical for an indoor news setting, creating soft lighting.*

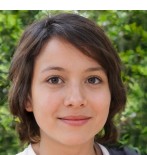

**Type:** Entire Face Synthesis

**Face:** *The person in the image is a young adult female with shoulder-length brown hair, a gentle expression, and light, expressive eyes, and is dressed in a white garment.*

**Environment:** *The image has an outdoor setting with green foliage in the background, possibly in a park or garden, with soft, natural lighting illuminating the face.*

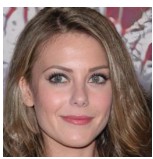

**Type:** Face Editing

**Face:** *The image shows a young woman with long, straight hair, wearing makeup and a slight smile on her face.*

**Environment:** *The background is blurred with abstract red and white designs, and the lighting is soft and evenly illuminates the face.*

Figure 7: Visualizations of annotated data. Each image in our unified dataset is labeled in two ways. First, we assign one of five types through human annotation. Then, we generate two distinct text descriptions related to facial attributes and environmental factors using VLM-based VQA.

Table 11: Text label for *REAL* type. The first simple text label, highlighted in **bold**, is used in the image-text classification evaluation. The remaining augmented labels provide a range of descriptive variations that enhance the model's understanding of *REAL* images.

| Set of Text Labels Assigned to $t_{REAL}$ |
| --- |
| ***"a photo of a real face."*** |
| *"a naturally captured image, preserving the individual's real identity."* |
| *"an untainted photograph showing the person in their true essence."* |
| *"a legitimate image capturing the subject's natural essence."* |
| *"a real and unedited portrayal of the subject's appearance."* |
| *"a natural image displaying the individual in their genuine state."* |
| *"a true photographic capture of the person in their real form."* |
| *"a completely original image capturing the real essence of the individual."* |
| *"a true photograph of the individual, free of any artificial influence."* |
| *"a true-to-life image portraying the individual as they are."* |
| *"an unmanipulated photo reflecting the subject's true identity."* |
| *"an unembellished image depicting the subject's true self."* |
| *"a raw image that genuinely reflects the person's appearance."* |
| *"a photo that faithfully represents the subject's true face."* |
| *"a legitimate image preserving the subject's true form."* |
| *"a naturally captured photograph preserving the person's identity."* |
| *"a true-to-life depiction of the subject as they exist."* |
| *"an unprocessed representation of the subject's natural form."* |
| *"a natural photo, free from the influence of any digital technology."* |
| *"a pristine image that genuinely reflects the subject's features."* |
| *"a truthful and unaltered photo of the subject's face."* |
| *"an unaltered photograph revealing the person's authentic appearance."* |
| *"a straightforward image emphasizing the individual's genuine identity."* |
| *"a straightforward, unedited photograph of the person."* |
| *"a clear, natural portrayal of the individual's features."* |
| *"an unaltered capture of the subject's genuine features."* |
| *"an unprocessed image portraying the subject as they naturally appear."* |
| *"a genuine photo capturing the true identity of the individual."* |
| *"a raw depiction of the subject's natural traits."* |
| *"a pure, unaltered depiction of the individual's face."* |

Table 12: Text label for *Face Swapping* type. The first simple text label, highlighted in **bold**, is used in the image-text classification evaluation. The remaining augmented labels provide a range of descriptive variations that enhance the model's understanding of *Face Swapping* images.

| Set of Text Labels Assigned to $t_{FS}$ |
| --- |
| ***"a photo of a face swapping."*** |
| *"a synthetic image where the face transition appears imperceptible."* |
| *"a manipulated image where one individual's face has been seamlessly replaced with another."* |
| *"a synthesized image featuring the flawless substitution of one face for another."* |
| *"an example of face transposition where two identities are seamlessly integrated."* |
| *"a seamless face overlay creating the illusion of identity transference."* |
| *"an edited photo blending two identities through advanced facial synthesis."* |
| *"a reimagined image where one person's face has been digitally replaced with another's."* |
| *"a realistic visualization of one face superimposed onto another individual."* |
| *"a deepfake alteration creating the illusion of the source face belonging to the target."* |
| *"a face-swapped image retaining the natural appearance of the overall photograph."* |
| *"a computational synthesis of one person's face overlaid onto another's body."* |
| *"a deepfake-crafted photo replacing one individual's face with another's."* |
| *"a deepfake showcasing the complete substitution of the original face."* |
| *"an altered image with a flawlessly transplanted face from another source."* |
| *"a face-swapped image designed to merge two identities into a convincing portrayal."* |
| *"an expertly replaced face that merges the identities of two individuals."* |
| *"a digitally altered face that replaces the original while retaining realism."* |
| *"a photo demonstrating the realistic substitution of one face for another."* |
| *"a synthetic rendering that replaces the original face with a new, realistic identity."* |
| *"a hybridized photo where the facial features of two people are merged."* |
| *"a realistic depiction of face replacement using cutting-edge AI algorithms."* |
| *"a digital transition of one face into another's context, creating a cohesive visual."* |
| *"a visually realistic replacement of one face with another using AI techniques."* |
| *"a creative deepfake, blending two distinct facial identities in one frame."* |
| *"a reconfigured image where the original identity has been replaced with a new face."* |
| *"a photograph that combines the facial identity of the source with the context of the target."* |
| *"a synthetic face replacement that mimics the appearance of another individual."* |
| *"an altered image with a flawlessly exchanged face, keeping the target's proportions."* |

Table 13: Text label for *Face Reenactment* type. The first simple text label, highlighted in **bold**, is used in the image-text classification evaluation. The remaining augmented labels provide a range of descriptive variations that enhance the model's understanding of *Face Reenactment* images.

| Set of Text Labels Assigned to $t_{FR}$ |
|---|
| ***"a photo of a face reenactment."*** |
| *"a dynamic synthesis combining source facial expressions with a target's identity."* |
| *"a manipulated image where source-driven behaviors redefine the target's facial animations."* |
| *"a synthetic face combining the target's features and the source's expressive cues."* |
| *"a dynamic facial synthesis that animates the target with the source's emotional attributes."* |
| *"a synthesized depiction of a target face animated by the source's emotional behavior."* |
| *"a reenacted face, maintaining the target's appearance while expressing the source's gestures."* |
| *"a manipulated target face enhanced with external expressive characteristics."* |
| *"a synthesized face showcasing transferred expressions while preserving the target's identity."* |
| *"a target face retaining its identity but animated by the source's expressive behavior."* |
| *"a synthetic animation that mimics the source's expressions on the target's features."* |
| *"a motion-transferred facial image, preserving the target's look but expressing the source's feelings."* |
| *"a facial synthesis driven by the source's emotive and dynamic behaviors."* |
| *"a synthesis of dynamic expressions from a source, applied to a target's visage."* |
| *"a synthetic reenactment combining the source's emotive characteristics with the target's identity."* |
| *"a face manipulation highlighting the dynamic interplay of source emotions on the target."* |
| *"a dynamically altered image where the expressions of a source person have been imposed on the target's face."* |
| *"a reenacted depiction where the source's emotional gestures guide the target's behavior."* |
| *"a manipulation that transfers source emotive behavior onto the target with realism."* |
| *"a synthesized animation, infusing source emotions into a preserved target identity."* |
| *"a synthetic reenactment image that transfers the essence of source movements."* |
| *"a reenactment image that synchronizes the source's emotional expression onto the target."* |
| *"a deepfake creation where the source's expressions animate the target's face."* |
| *"a reenactment-based image, preserving the target's static identity while adding source movements."* |
| *"a reenacted target face enriched with the emotive characteristics of the source individual."* |

Table 14: Text label for *Entire Face Synthesis* type. The first simple text label, highlighted in **bold**, is used in the image-text classification evaluation. The remaining augmented labels provide a range of descriptive variations that enhance the model's understanding of *Entire Face Synthesis* images.

| Set of Text Labels Assigned to $t_{EFS}$ |
| --- |
| ***"a photo of a entire face synthesis."*** |
| *"a digital portrayal of a human face, generated entirely by deep learning."* |
| *"an artificial human visage created with advanced machine learning tools."* |
| *"an ai-generated facial image designed to mimic natural human appearance."* |
| *"a generative model's output of a human face, appearing lifelike but not real."* |
| *"a completely synthesized human face, crafted using deep learning algorithms."* |
| *"a simulated human face created by neural networks to appear authentic."* |
| *"an artificial face image generated to exhibit human-like detail and emotion."* |
| *"a synthetic facial image where every element is the product of ai model."* |
| *"a representation of a human face that is a digital construct, not a real person."* |
| *"a high-resolution image of a face that is completely ai-generated."* |
| *"a face generated from a neural network trained to create photorealistic human features."* |
| *"a fully artificial human face crafted by advanced ai technologies."* |
| *"a non-real facial image created entirely through ai-driven synthesis techniques."* |
| *"a photorealistic face that showcases ai's ability to generate human-like features."* |
| *"a digital face image, where every aspect is synthesized."* |
| *"an example of ai-driven facial synthesis, where the face is purely artificial."* |
| *"an ai-crafted face, where realism is achieved through advanced neural techniques."* |
| *"a digital creation of a face, formed entirely by deep learning synthesis."* |
| *"a neural network-generated face that appears entirely plausible."* |
| *"a computer-generated human face that mirrors the natural variation of real individuals."* |
| *"a fabricated face image, where every detail is the result of neural network generation."* |

Table 15: Text label for *Face Editing* type. The first simple text label, highlighted in **bold**, is used in the image-text classification evaluation. The remaining augmented labels provide a range of descriptive variations that enhance the model's understanding of *Face Editing* images.

| Set of Text Labels Assigned to $t_{FE}$ |
| --- |
| ***"a photo of a face editing."*** |
| *"a deepfake face with modified visual characteristics while preserving the original identity."* |
| *"an altered face image with adjustments to specific attributes such as age and gender."* |
| *"a synthetically enhanced face with refined or transformed facial features."* |
| *"an edited portrait where certain aspects of the face have been subtly or significantly changed."* |
| *"a manipulated face that showcases alterations in appearance without changing the core identity."* |
| *"a modified facial image that presents adjusted traits while maintaining recognizable features."* |
| *"a deepfake-generated face with enhanced or altered attribute details."* |
| *"an ai-edited face image focusing on the transformation of facial characteristics."* |
| *"a synthetically modified face exhibiting adjusted features while retaining its essence."* |
| *"a digitally enhanced face displaying changes to key appearance attributes."* |
| *"an adjusted portrait that demonstrates transformations in facial aspects while remaining authentic."* |
| *"a modified face image that retains overall identity while refining selected attributes."* |
| *"an altered image where specific facial details have been modified to achieve a desired look."* |
| *"a manipulated portrait with tailored changes to improve or transform visual features."* |
| *"an ai-processed face displaying modified traits while keeping the original structure intact."* |
| *"a face image where certain attributes have been altered while preserving overall facial identity."* |
| *"an edited facial image that reflects changes in specific visual features without affecting recognition."* |
| *"a modified portrait that presents an adjusted version of the original face."* |
| *"a transformed facial image that showcases refinements in appearance while maintaining a natural look."* |
| *"an adjusted face that highlights changes in selected attributes while retaining key facial features."* |
| *"a synthetically altered portrait where facial details have been subtly enhanced or modified."* |
| *"an ai-edited face image with visible improvements or changes to certain characteristics."* |
| *"a deepfake face showing controlled modifications in appearance while keeping identity consistent."* |
| *"a processed face that reflects tailored adjustments to achieve a refined appearance."* |
| *"a modified image where selective facial attributes have been enhanced or transformed."* |
| *"an edited portrait that demonstrates a balance between original structure and refined details."* |
| *"a transformed face where visual attributes have been modified while maintaining realism."* |

