# OpenReview forum: "A Rich Knowledge Space for Scalable Deepfake Detection"
_ICLR.cc/2026/Conference — ICLR 2026 Poster_

### Official Review · Reviewer_cYdy · 2025-10-29

**Soundness:** 3
**Presentation:** 3
**Contribution:** 3
**Rating:** 6
**Confidence:** 4

**Summary:**

This paper introduces SD2 (Scalable Deepfake Detection), a CLIP-based visual-language model trained on a newly built MMI-DD dataset containing 3.6 million multimodal facial images across 11 datasets. It proposes three main components: the Cross-Layer Attention Module (CLAM), the Text Label Separation Loss, and the Dual Contrastive Loss to enhance generalization and scalability in deepfake detection. Experimental results demonstrate strong performance across multiple benchmarks, suggesting that SD2 scales effectively with increasing data size.

**Strengths:**

1. The authors integrate eleven existing datasets into a unified corpus with over 3.6 million multimodal samples, covering four manipulation types and both real and fake instances (Section 3.1, Table 1). They also include human-verified textual labels and VLM-generated descriptions, enhancing the dataset’s semantic richness and enabling cross-modal learning (Figure 2). This large-scale integration and annotation pipeline offer a solid infrastructure that benefits future research in multimodal forensics.

2. The model introduces the Cross-Layer Attention Module (CLAM) to fuse visual features from different transformer layers and employs Dual Contrastive and Label Separation losses to strengthen cross-modal alignment. Each component is separately evaluated in the ablation study, where removing CLAM or either loss leads to performance drops of 1.24–1.97% on the cross-domain mAUC. This structured experimental setup provides convincing evidence that the proposed components meaningfully contribute to the final performance. Across intra-domain and cross-domain tests, SD2 achieves 95.76 mAUC and 87.79 mAUC respectively, surpassing prior models and methods. What I like is that the authors further report that performance continues to improve as training data size increases from 1M to 3.6M samples (Table 6). These findings support the paper’s claim that SD2 scales effectively and avoids the overfitting observed in earlier CLIP-based detection frameworks.

3. The extension to non-facial AIGC data highlights broader applicability. The experiments on non-facial synthetic datasets, such as synthetic scene and object generation benchmarks, show that SD2’s vision encoder generalizes beyond facial forgeries. The model outperforms prior works on the averaged mAUC, confirming the potential of multimodal pretraining for generalized forgery detection. This result is a noteworthy attempt to bridge the gap between face-centric and content-agnostic AIGC detection, which is important in the current stage.

**Weaknesses:**

1. The overall novelty of SD2 is somehow limited, as most techniques extend existing designs rather than introduce new concepts. The proposed Cross-Layer Attention Module (CLAM) aggregates multi-level visual features from intermediate transformer layers. However, similar multi-scale fusion strategies have long existed in CNN-based architectures, and recent ViT variants also exploit cross-layer interactions. As a result, CLAM feels more like a standard adaptation of established ideas to a CLIP backbone rather than a novel architectural contribution.

2. Parts of the loss terms largely reuse ideas already present in prior multimodal learning frameworks. For example, the Dual Contrastive Loss directly follows CLIP-style image-text alignment, while the Text Label Separation Loss resembles class-wise embedding decorrelation objectives used in prior works.  Without further explanations and comparisons, the learning objective feels incremental rather than innovative.

3. Certain implementation details remain unclear, for example,  parameter updates in Eq. (4). The paper introduces the Text Label Separation Loss (Section 3.3, Eq. 4) but I can not find out which parts of the model are updated under this loss. Clarifying which components receive gradient updates would  strengthen the technical clarity of the paper.

**Questions:**

Please check Weaknesses for the details.

---

> ### Author Response · Authors · 2025-11-25
>
> > We are grateful for the positive evaluation of our work. In particular, we are pleased that the reviewer recognized the value of our scalability analysis observed from large-scale data. We address the questions raised by the reviewer in the following response.
> ***
> ### **1. Rationale for CLAM Design in Large-Scale Detection (W1)**
> We appreciate the reviewer’s comments on multi-scale fusion. We acknowledge that the principle of aggregating features across network layers is an established concept.
>
> * **Methodological Validation at Scale**
> However, we emphasize that the core contribution of CLAM lies in its methodological and empirical validation specifically for the large-scale deepfake detection domain. Due to the limited exploration of deepfake detection at this scale, there is a scarcity of evidence on whether multi-scale fusion strategies remain effective or how they should be optimally designed.
>
> * **Alignment with Research Trends**
> Furthermore, as noted in our paper (Sec. 2.2), recent research trends prioritize the effective adaptation of Foundation Models (e.g., CLIP) over designing entirely novel architectures.
>
> * **Synergy with Text Label Separation Loss**
> Our research aims to train and detect synthetic images across diverse manipulations at a large scale. As demonstrated in prior works [1, 2], each manipulation method leaves its own unique forensic artifacts. Unlike conventional binary approaches, our Text Label Separation Loss ($\mathcal{L_S}$) is explicitly designed to optimize the separation of four fine-grained manipulation types. We hypothesized that CLAM would create a strong synergy with $\mathcal{L_S}$ by capturing the distinct low-level forensic cues specific to each type. Aligning with this hypothesis, the integration of CLAM with $\mathcal{L_S}$ effectively contributes to performance improvement, as demonstrated in Table 5 (+1.6% mAUC).
>
> **Therefore, while CLAM builds on established ideas in feature fusion, we believe its value lies in providing essential insights into this under-explored research area and offering a systematically optimized design that synergizes with our specific learning objectives**.
> ***
> ### **2. Rationale for Dual Contrastive Loss and Text Label Separation Loss (W2)**
> We appreciate the reviewer’s observation regarding the familiarity of individual loss components. While Dual Contrastive Loss ($\mathcal{L_D}$) and Text Label Separation Loss ($\mathcal{L_S}$) objectives have appeared in prior works, our contribution lies in their specific adaptation and integration for large-scale forensic vision-language modeling.
>
> * **Addressing Embedding Collapse in Native CLIP Space**
> A key challenge, not addressed by prior works, is that image-text contrastive alignment alone cannot yield reliable classification because the five forensic text labels remain insufficiently separated in the native CLIP embedding space (See Figure 5 in revised version). Without explicit separation, images aligned to these labels collapse into a highly entangled region. **Our $\mathcal{L_S}$ provides the necessary structure** by enforcing clear distinctions among the five label embeddings, thereby supporting the contrastive objective and enabling effective classification.
>
> * **Mitigating Spurious Correlations via $\mathcal{L_D}$**
> A recent study [3] demonstrates that standard CLIP objectives are inherently vulnerable to spurious correlations (e.g., relying on background context rather than the object itself). We identify a parallel vulnerability in deepfake detection, where models often rely on scene context rather than forensic traces. This vulnerability motivated us to introduce $\mathcal{L_D}$, which explicitly disentangles facial cues from environmental context to ensure robust forensic learning.
>
> As noted in the reviewer’s strengths section, **each component is rigorously evaluated in our ablation study, demonstrating that their contributions are indeed complementary**.

---

> ### Author Response · Authors · 2025-11-26
>
> ### **3. Text Separation Loss - Update parameters (W3)**
> The purpose of our Text Label Separation Loss ($\mathcal{L_S}$) is to enhance the discriminability among our constructed text labels, which tend to remain insufficiently separated in the native CLIP text embedding space (See Figure 5 in revised version). By enforcing the similarity matrix to approximate the identity matrix, this loss drives clearer separation among the five text label types.
>
> Regarding the implementation details, since this objective optimizes the text embeddings, **the gradients derived from $\mathcal{L_S}$ are back-propagated exclusively to the Text Encoder.** Specifically, we update the encoder through the LoRA fine-tuning mechanism. **Importantly, only the LoRA parameters attached to the Text Encoder are updated, while the pre-trained CLIP text encoder weights remain frozen.**
>
> ***
> REFERENCES
> [1] Sheng-Yu Wang, et al. "CNN-generated images are surprisingly easy to spot... for now." In *Proceedings of the IEEE/CVF conference on computer vision and pattern recognition (CVPR)*, 2020.
> [2] Riccardo Corvi, et al. "On the detection of synthetic images generated by diffusion models." In *IEEE International Conference on Acoustics, Speech and Signal Processing (ICASSP)*, 2023.
> [3] Qizhou Wang, et al. "A sober look at the robustness of clips to spurious features." In *The 38th Conference on Neural Information Processing Systems (NeurIPS)*, 2024.
>
> > We sincerely appreciate the reviewer's thorough examination and deep understanding of our objective functions. We agree that individual components leverage established strategies;
> however, we have clearly articulated the rationale for introducing each component and systematically unified them into a single framework that yields high synergy in a complementary manner for large-scale deepfake detection. We believe these contributions merit a positive re-evaluation.

---

> ### Comment · Area_Chair_xFnc · 2025-11-28
>
> Dear Reviewers,
>
> Thank you for your time and thoughtful feedback on this manuscript.
>
> The authors have now submitted their rebuttal. If you haven’t already, we kindly ask you to review their responses and consider whether your concerns have been adequately addressed.
>
> Best regards,
>
> AC

---

### Official Review · Reviewer_zo7x · 2025-10-30

**Soundness:** 4
**Presentation:** 3
**Contribution:** 3
**Rating:** 6
**Confidence:** 5

**Summary:**

The paper finetunes CLIP image and text encoder with image-text deepfake pairs by curating all the existing datasets. The text and caltegorical annotation for each image is generated using InternVL and gpt o1 respectively.

**Strengths:**

1. The paper is well motivated. I agree that the current deepfake methods in the community train on FF++ and test on other datasets which doesn't necessarily reflect their in-the-wild performance and are overfitted to the training domain. The proposed methods aims to scale the data bottleneck by training in a contrastive manner using image-text pairs, ands highlight that the model performance improves with dataset size.
2. The paper is well-written.
3. The performance improvement with data scaling is impressive.
4. The data scaling shows emergent behavior and the performance improves on datasets "beyond facial manipulations".

**Weaknesses:**

The paper doesn't compare with recent deepfake detection methods. The authors should compare the recent methods by training them using the bigger curated dataset proposed, and then compare it with SD^2, to see whether there is a performance improvement or not . This will highlight whether the CLIP-based LoRA training is actually useful or it is just because of data scaling where the CLIP-based methods are not scalable. It is still not clear that other non-CLIP based methods are scalable or not.

Please compare with 1 or 2 SOTA methods from NeurIPS 25, CVPR 25, AAAI 25, etc.

**Questions:**

1. Can you show the results of recent methods when trained on the bigger dataset and show the performance comparison.

---

> ### Author Response · Authors · 2025-11-25
>
> > We appreciate the reviewer’s positive assessment, especially the high evaluation of our work's contribution to scalability in deepfake detection. We provide detailed responses to the additional questions below.
> ***
> ### **1. SOTA Comparison and Uniform Setting (W1, Q1)**
> We are pleased to receive the reviewer’s keen interest in our scalability investigation of existing deepfake detection research. While we appreciate the additional question regarding this aspect, **we would like to respectfully clarify that our study already addresses the requested points**
>
> * **Identical Setting**
> We emphasize that all comparisons (Table 2, 3, 5, and 6) were conducted under strictly identical training data and conditions. Specifically, the experiments in Table 6 constitute a controlled scale-variation study, directly demonstrating the scalability of each method under identical settings.
>
> * **Scalability: CLIP-based vs. Non-CLIP Approaches**
> Our scalability analysis (Table 6) directly shows that **all baselines fail to achieve scalable detection.** Non-CLIP methods (Xception, RECCE, SPSL) exhibit early saturation. Similarly, strong CLIP adaptations like CLIP-LoRA and CLIP-SVD also plateau or degrade in performance. This demonstrates that **only our $\mathcal{SD}^{2}$ framework exhibits consistent performance gains.**
>
> * **Comparison with Recent SOTA methods**
> We would like to clarify that SOTA methods were already included. Specifically, CLIP-SVD (Effort) [1] is a state-of-the-art method published in ICML 2025, which is later than CVPR 2025. This direct comparison addresses the request for benchmarking against the most recent methods.
>
> *We hope that our response fully addresses your concerns*.
> ***
> References
> [1] Zhiyuan Yan, et al. "Orthogonal Subspace Decomposition for Generalizable AI-Generated Image Detection." In *Forty-second International Conference on Machine Learning (ICML)*, 2025

---

> ### Comment · Area_Chair_xFnc · 2025-11-28
>
> Dear Reviewers,
>
> Thank you for your time and thoughtful feedback on this manuscript.
>
> The authors have now submitted their rebuttal. If you haven’t already, we kindly ask you to review their responses and consider whether your concerns have been adequately addressed.
>
> Best regards,
>
> AC

---

### Official Review · Reviewer_6zHj · 2025-10-31

**Soundness:** 2
**Presentation:** 2
**Contribution:** 2
**Rating:** 6
**Confidence:** 5

**Summary:**

- This paper presents a visual–language framework for deepfake detection together with a large-scale, multi-modal knowledge space constructed by unifying 11 datasets into a 3.6M-image corpus with consistent preprocessing, five-way type labels (REAL/FS/FR/EFS/FE), and VLM-generated facial and scene descriptions. The goal is to achieve scalable, cross-domain generalization beyond single-dataset training by leveraging richer supervision from text and multi-level visual cues. Specifically, the method (SD2) adapts CLIP with a cross-layer attention module that fuses intermediate and final vision features, a fine-grained image–text classification objective using type-specific labels, and a text-label separation loss to stabilize class embeddings, together with a dual image–text contrastive loss that aligns each image with both facial and scene descriptions. To enable efficient and robust training at scale, the approach fine-tunes CLIP encoders with LoRA and employs a SigLIP-style contrastive objective to avoid very large batches. Furthermore, extensive experiments show superior intra-domain and cross-domain AUC over CNN/Transformer and CLIP-based baselines, and competitive accuracy on a non-facial AIGC benchmark using the frozen vision encoder with linear probing, underscoring the method’s effectiveness, scalability, and practical applicability.

**Strengths:**

- Overall, the paper is clearly written and easy to follow, with a well-articulated connection between the unified corpus and the proposed vision–language (V–L) objectives.
- The data contribution is significant: the authors unify 11 sources into a 3.6M-image corpus with consistent preprocessing, five-way labels, and VLM-generated descriptions.
- The proposed method is both practical and modular, building on CLIP by introducing cross-layer attention, text–label separation, and dual image–text contrastive training.
- The authors also address training efficiency, employing LoRA fine-tuning and a SigLIP-style objective to reduce batch requirements. The results are strong across both intra- and cross-domain settings, with additional linear-probe transfer experiments on a non-facial AIGC benchmark further demonstrating robustness.

**Weaknesses:**

- The unified corpus may inherit dataset biases and spurious correlations; issues such as deduplication, identity overlap, and source-specific artifacts are not sufficiently quantified.
- VLM-generated descriptions can introduce noise or hallucinations, yet the paper does not assess caption quality or analyze its impact on downstream accuracy.
- The reported performance gains may be partly attributable to corpus scale or data curation rather than architectural improvements; stronger experimental controls, e.g., equalizing data size, testing alternative prompts, or using simpler adapters.

**Questions:**

- How are near-duplicates and identity overlaps managed across the 11 sources? Could clarify the criteria for duplicate removal, report per-source identity counts, and describe any identity-disjoint checks implemented to prevent data leakage.
- How accurate are the VLM-generated facial and scene descriptions? Have authors conducted a small-scale human audit (e.g., inter-annotator agreement, error type analysis) or performed an ablation study in which captions are shuffled or masked to quantify their causal impact?
- How are heterogeneous source labels mapped to the five target categories (REAL/FS/FR/EFS/FE)? Does proposed framework support open-set or “other manipulations” (e.g., audio-visual inconsistencies, partial edits, diffusion-based reenactment), and how are such cases handled?
- How sensitive are the results to prompt templates, temperature, and LoRA rank choices? Report the random seeds, hyperparameter ranges, and effect sizes (with confidence intervals) for main comparisons.

**Details Of Ethics Concerns:**

- The paper integrates numerous face-forensics datasets into a new corpus (MMI-DD) and plans to release it with added VLM-generated captions and unified labels. Given cross-jurisdiction issues (e.g., GDPR/CCPA), I flag for review on privacy, security, and legal compliance before data release.

---

> ### Author Response · Authors · 2025-11-25
>
> > We appreciate the reviewer’s detailed assessment and the recognition of our significant data contribution and the practical nature of the proposed framework. We address each point below.
> ***
> ### **1. Quantification of Identity Overlap and Data leakage (W1, Q1)**
> We report the requested Identity Counts and Sources in Table A. While exact overlap ratios cannot be quantified due to missing identity metadata, we explain below how we mitigated overlaps and prevented data leakage.
>
> * **Managing Identity Overlaps with Quality Variations (same source, different quality)**
> For pairs such as DF40 & FF++ (C40), TIMIT (LQ) & TIMIT (HQ), and CelebA & CelebA-HQ, the datasets share the same source identities but differ significantly in visual quality.
>
>   **(1) Benefit:** Instead of redundancy, we consider these variations as a form of data augmentation. By exposing the model to different quality levels, we enhance its robustness against compression and resolution issues.
>   **(2) Leakage Prevention:** To prevent data leakage, we followed the official train/test splits for datasets like FF++ and TIMIT. For CelebA and CelebA-HQ, we used their shared image IDs to split the data, preventing data leakage.
>
> * **Managing Identity Overlaps within DFFD (same source, same quality)**
> The real data in DFFD originates from FFHQ, CelebA, and FaceForensics++.  Crucially, as these images maintain the same source and quality, a rigorous filtering and processing pipeline is required.
>
>   **(1) Deduplication:** To avoid redundancy, we removed the FaceForensics++ subset from DFFD before merging it into our corpus, MMI-DD. This clarifies why our reported DFFD count (20,000) in Table 1 is smaller than the original statistics (58,703).
>   **(2) Pre-compression Strategy:** For the remaining subsets (FFHQ and CelebA), we applied pre-compression (JPEG quality 60-100) prior to storage. We then generated VLM-based text descriptions from these pre-compressed images. We intentionally introduced compression processing to mimic the relationship between DF40 (High Quality) and FF++ (Low Quality).
>
> Table A. Identity counts and dataset sources
> | Index | Dataset Name | Identity Count (Subjects) | Dataset Source |
> | :---: | :--- | :--- | :--- |
> | 1 | **DF40** | 1,059 | FaceForensics++, CDF |
> | 2 | **DFF** | Unknown | IMDb-WIKI |
> | 3 | **DFFD** | Unknown | FFHQ, CelebA, FaceForensics++ |
> | 4 | **DFDCP** | 66 | Paid Actors |
> | 5 | **KoDF** | 403 | Paid Korean Subjects |
> | 6 | **FF++(C40)** | 1,000 | YouTube Videos |
> | 7 | **TIMIT(LQ)** | 32 | VidTIMIT |
> | 8 | **TIMIT(HQ)** | 32 | VidTIMIT |
> | 9 | **CelebA** | 10,177 | Web Crawled |
> | 10 | **CelebA-HQ** | Unknown | CelebA |
> | 11 | **FFHQ** | 70,000 | Flickr |
>
> *We hope the reviewer's concerns are resolved with our explanation. We have incorporated these details and the table A into the revised manuscript (See Section 3.1).*

---

> > ### Author Response · Authors · 2025-11-25
> >
> > ### **2. VLM-generated facial and environmental description (W2, Q2):**
> > We thank the reviewer for raising this valid concern regarding caption reliability. While specific accuracy metrics are not included, we substantiate the high reliability of our captions by referencing related works that report high accuracy in identical research contexts. Furthermore, we quantitatively demonstrate that our descriptions, validated through internal verification, yield significant performance improvements.
> >
> > * **Annotation using MLLM**
> > If we were employing the MLLM for forensic analysis (e.g., extracting forensic cues), a rigorous verification process might be necessary, as seen in studies [1, 2, 3, 4]. However, our research uses MLLM (InternVL 2.5) only to extract **global semantic information** (facial and environmental attributes). As the reviewer noted, **this is precisely what MLLMs excel at**. Aligning with this recognition, recent literature has increasingly leveraged MLLMs for semantic annotation [5, 6].
> >
> > * **Empirical Evidence - High Reliability of Our Annotations**
> > Furthermore, recent experimental findings [7] strongly support our approach. This study specifically investigated the **reliability of VLMs for facial attribute annotation (on the CelebA dataset)** and demonstrated that SOTA VLMs achieve **89.1% consistency** with verified human annotations. This quantitative evidence directly refutes the concern that such annotations are 'highly unreliable.' The study further notes that **the remaining discrepancies often stem from inherent subjectivity** (e.g., hair shade, attractive eyes, big nose) rather than hallucinations, confirming that the semantic signal provided by VLMs is robust and trustworthy for our purpose.
> >
> > * **Internal Verification & Demonstrated Effectiveness**
> > To provide stronger internal verification, we manually audited randomly sampled images from each dataset and confirmed that the generated descriptions align with the corresponding visual content. We demonstrate our approach’s effectiveness quantitatively (Tab. 3) under a rigorous and realistic evaluation protocol, adopting seven unseen datasets. Moreover, the ablation study (Table 5) shows that both the Facial and Environmental descriptions consistently improve performance.
> >
> > *We have included these details into Section 3.2 of the revised manuscript*
> > ***
> > ### **3. How to Predefine Fine-Grained Fake Type (Q3)**
> >
> > The reviewer raises an important point regarding how fine-grained manipulation types are defined and mapped. We fully agree that it is the crucial point for our framework, and **we clarify below that our approach remains flexible**.
> >
> > Our five-category supervision (REAL/FS/FR/EFS/FE) adheres to the standard forensic typology established by prior surveys [8] and recent benchmarks [9].
> >
> > * **Extensibility to New Manipulation Methods**
> > While we recognize the emergence of novel manipulation methods (e.g., diffusion-based reenactment or partial edits), our framework is specifically engineered for extensibility. $\mathcal{SD}^2$ is not constrained by the current fake type taxonomy. New or emerging manipulations can be easily **incorporated by defining a new fine-grained category** (e.g., DIFFUSION-REENACTMENT class) and updating the associated semantic descriptions. Crucially, this expansion requires no change to our core learning pipeline.
> >
> > * **Binary classification**
> > As we stated in Sec. 5.1, $\mathcal{SD}^2$ operates as a binary classifier (REAL vs. FAKE decision) in inference time. Since the detector operates on general forensic cues rather than relying on any single predefined type, it effectively generalizes to novel manipulation methods during inference.
> >
> > As $\mathcal{SD}^2$ is designed as a specialized image-based detector, it naturally targets static visual artifacts. Therefore, the detection of audio–visual inconsistencies falls outside the scope of our research.

---

> ### Author Response · Authors · 2025-11-25
>
> ### **4. Hyperparameter Sensitivity (Q4)**
>
> The reviewer requested additional experiments to verify the robustness of our framework. We respectfully emphasize that large-scale training regimes inherently exhibit lower performance variation, and our framework consistently demonstrates this stability. As requested, we provide the detailed analysis below:
> * **Robustness to Prompt Templates**
> Our Separation Loss ($L_S$) effectively separates the prompts of the five types, while maximizing similarity among prompts of the same type. As shown in Figures 5 and 6 of the Appendix (revised version), intra-type text embeddings converge to a similarity of approximately 1.0, while inter-type embeddings become nearly orthogonal. **Consequently, diverse prompt variations yield consistent outputs.** We chose the simplest template solely for clarity, not because the performance was sensitive to it.
>
> * **Temperature setup**
> As noted in Sec. 4.2, we follow CLIP’s original temperature. Since our model is initialized from CLIP and fine-tuned, **modifying the temperature early disrupts the pretrained feature scale and leads to unstable optimization**.
>
> * **Robustness to LoRA Rank**
> We conducted additional experiments to assess the model's sensitivity to LoRA rank variations ($r_\text{lora} \in \{4, 8, 16\}$). To facilitate rapid experimentation, we utilized a representative 1M subset of MMI-DD for training. For evaluation, we adhered to the rigorous protocol used in the main paper (Table 5 and 6), reporting mAUC across seven unseen datasets. Table B demonstrates that our model is robust to rank variations.
>
> * **Seeds, Hyperparameter Ranges, Effect Sizes**
> We further investigate the impact of the hyperparameter $\alpha$ in Eq. (6). Using a setup identical to the LoRA rank experiments, Table C shows that our method remains robust across different $\alpha$ values.
>
> While we kindly ask for more details on the expected 'Seeds' (e.g., specific seed values vs. variance analysis) and 'Effect sizes,' **we emphasize that Table B and C already provide strong evidence of our framework's robustness against varying experimental configurations.**
>
> Table B. Effect of LoRA rank variations on cross-domain mAUC performance.
> | $r_\text{lora}$ | $\alpha_\text{lora}$ | mAUC |
> | :---: | :---: | :---: |
> | 4 | 32 | 82.97 |
> | **8** | **32** | **84.85** |
> | 16 | 32 | 84.37 |
>
> Table C. Effect of hyperparameter $\alpha$ values on cross-domain mAUC performance.
> | α | mAUC |
> | :---: | :---: |
> | 1.5 | 83.45 |
> | **2.0** | **84.85** |
> | 2.5 | 83.02 |
>
> *We have incorporated the requested experimental results into Section 5.3 of the revised manuscript.*
> ***
>
> ### **5. Validity of Performance Gains (W3)**
>
> The reviewer raises a concern that our performance gains might stem from unfair experimental conditions. **However, we emphasize that all comparisons (Table 2,3,5, and 6) were conducted under strictly identical training data and conditions.** Notably, the experiments (Table 3, 5, and 6) were conducted under a rigorous setting on seven unseen datasets, where our method consistently achieves superior results.
>
> * **Ablation study: Architectural Benefits**
> To isolate architectural contributions, we analyzed the impact of each key component (Table 5). Every component in our framework contributes to the performance. Notably, even using only $\mathcal{L}_\text{C}$ and CLAM, our model already surpasses CLIP-LoRA (81.84 vs. 72.12).
>
> * **Ablation study: scalability**
> To demonstrate scalability, we ran controlled scale-variation experiments (Table 6) under an identical training environment for all baselines. Our framework maintains superior performance across all training data sizes (from 100K to 3M). Our superior performance does not originate from different training data scale.
>
> **Overall, our model demonstrates strong architectural benefits and superior performance across all data sizes under rigorous and identical settings.** Additionally, as discussed in *4. Hyperparameter sensitivity (Q4)*, we theoretically clarified that our prompts yield consistent outputs.
>
> *We hope these clarifications resolve the reviewer's concerns and encourage a positive re-evaluation of our work.*

---

> ### Author Response · Authors · 2025-11-25
>
> REFERENCES
> [1] Yue Zhang, et al. "Common sense reasoning for deepfake detection." *European conference on computer vision*. Cham: Springer Nature Switzerland, 2024.
> [2] Ziyin Zhou, et al. "AIGI-Holmes: Towards Explainable and Generalizable AI-Generated Image Detection via Multimodal Large Language Models." In *Proceedings of the IEEE/CVF International Conference on Computer Vision (ICCV)*, 2025.
> [3] Siwei Wen, et al. "Spot the fake: Large multimodal model-based synthetic image detection with artifact explanation." In *Proceedings of the 39th Conference on Neural Information Processing Systems (NeurIPS)*, 2025.
> [4] Hengrui Kang, et al. "LEGION: Learning to ground and explain for synthetic image detection."  In *Proceedings of the IEEE/CVF International Conference on Computer Vision (ICCV)*, 2025.
> [5] Xudong Wang, et al. "TF-FAS: twofold-element fine-grained semantic guidance for generalizable face anti-spoofing." European Conference on Computer Vision. Cham: Springer Nature Switzerland, 2024.
> [6] Sachit Menon and Carl Vondrick. "Visual classification via description from large language models." In *The Eleventh International Conference on Learning Representations (ICLR)*, 2023.
> [7] Haoming Lu and Feifei Zhong. "Can Vision-Language Models Replace Human Annotators: A Case Study with CelebA Dataset." In *The Thirty-Eighth Annual Conference on Neural Information Processing Systems Workshop (EvalEval)*, 2024.
> [8] Zhiyuan Yan, et al. "Df40: Toward next-generation deepfake detection." In *The 38th Conference on Neural Information Processing Systems (NeurIPS)*, 2024.
> [9] Yisroel Mirsky and Wenke Lee. "The creation and detection of deepfakes: A survey." *ACM computing surveys (CSUR)* 54.1 (2021): 1-41.

---

> ### Comment · Area_Chair_xFnc · 2025-11-28
>
> Dear Reviewers,
>
> Thank you for your time and thoughtful feedback on this manuscript.
>
> The authors have now submitted their rebuttal. If you haven’t already, we kindly ask you to review their responses and consider whether your concerns have been adequately addressed.
>
> Best regards,
>
> AC

---

### Official Review · Reviewer_MC2Z · 2025-11-01

**Soundness:** 1
**Presentation:** 2
**Contribution:** 1
**Rating:** 0
**Confidence:** 5

**Summary:**

This paper introduces a new large-scale deepfake dataset called MMI-DD, which consolidates 3.6 million images from 11 different sources annotated with fine-grained DeepFake types. The data also contains VLM-generated text descriptions of facial and environmental attributes.​ The authors also propose SD², a scalable, CLIP-based detection baseline. The model uses a multi-modal learning strategy that leverages the detailed annotations through specialized classification and contrastive loss functions.

**Strengths:**

- The authors make a noteworthy attempt to address a significant challenge in the field: the lack of a large-scale, unified benchmark for DeepFake detection and reasoning. By integrating 11 distinct datasets into a single resource (MMI-DD), they provide a valuable foundation that could encourage the community to move beyond single-dataset training paradigms.
- The work highlights a critical and often overlooked issue, which is the performance saturation or degradation of existing models when trained on increasingly large and heterogeneous datasets.

**Weaknesses:**

- The paper states that six researchers manually categorized the images, but it does not report the inter-annotator agreement score or detail the protocol for resolving disagreements. This omission makes it difficult to assess the reliability and consistency of the fine-grained labels, which are crucial for the proposed classification loss.
- The annotations are generated using proprietary models like GPT-4o. This assumes that GPT-4o is a good DeepFake detector and reasoning model, which is wrong. MLLMs are inherently built for global semantics rather than capturing subtle inconsistencies that occur in DeepFakes. Papers like [1, 2] show that subtle changes are ignored, and these models are broadly good in overall content understanding. The paper also provides no information on how the quality, accuracy, and potential biases of these machine-generated captions were validated. This means that the annotations are highly unreliable and noisy.
- The paper makes claims about how its components work, for example, that CLAM captures low-level forgery artifacts. However, there is no qualitative evidence, such as attention maps or feature visualizations, to verify these claims and provide insight into the model's decision-making process.
- Section 5.2.2 states the model's vision encoder is frozen for linear probing but also mentions that the protocol involves fine-tuning on new data (SD v1.4 and ImageNet). This contradiction makes it difficult to determine if the strong performance is due to the pre-training on the MMI-DD dataset or the subsequent task-specific tuning.

[1] Tong, Shengbang, et al. "Eyes wide shut? exploring the visual shortcomings of multimodal llms." Proceedings of the IEEE/CVF Conference on Computer Vision and Pattern Recognition. 2024. [2] Huynh, Ngoc Dung, et al. "Vision-Language Models Can't See the Obvious." Proceedings of the IEEE/CVF International Conference on Computer Vision. 2025.

**Questions:**

- What procedure was followed to resolve labeling disagreements among the researchers?
- How did the authors verify the factual accuracy of the VLM-generated facial and environmental descriptions? Was there a human review process to filter out hallucinations or irrelevant details?
- What is the rationale behind the claim that the model is able to ground the reasoning to the visual cues through the CLAM module?
- The dual contrastive loss aims to disentangle forensic cues from spurious correlations. Have you conducted experiments to test this, for example, by evaluating the model on images where a known fake face is placed in diverse and unseen environments?
- In the general synthetic image detection experiment (Table 4), could the authors clarify whether the performance gains come from the MMI-DD pre-training or from the subsequent fine-tuning on ImageNet and SD v1.4 data?
- Figure 1 and Table 6 show that the proposed model scales better than baselines. How does the computational cost (e.g., total GPU hours) of training SD2 on the full 3.6 million image dataset compare to the other methods? And what are the inference times of each of these models?

---

> ### Author Response · Authors · 2025-11-25
>
> > We sincerely thank the reviewer for their thoughtful and constructive comments and questions. These points are invaluable to improve the quality of our work. Here, we address all the concerns raised by the reviewer and provide the details. In particular, we respectfully disagree with some of the points raised by the reviewer due to possible misunderstandings caused in the initial version. And, we clarified them, and provided the updated draft to address your concerns. Thanks again for the constructive comments!
> ***
> ### **1. Process of Human Annotation in MMI-DD (W1, Q1)**
> We would like to clarify that this annotation (4 fake types) was not a subjective, manual labeling of individual data points, but rather an objective, Generation method-level annotation from creators (authors). **The criteria for this mapping were clear, ensuring accurate processing.**
>
> * **Example of the annotation process**
> For instance, we investigated all manipulation techniques used in the KoDF dataset and confirmed that it contains subsets generated by six distinct generation methods (e.g., FaceSwap, DeepFaceLab, FSGAN, FOMM, etc.). We then categorized these methods. Since DeepFaceLab [1] is a technique that swaps facial identity, its entire subset was objectively annotated as 'FS' (as analyzed in our paper, DeepFaceLab is a 'Face Swapping' technique). We applied this process to all 57 generation techniques identified across the 11 integrated datasets, mapping each technique to one of our four deepfake types.
>
> *We hope this explanation fully resolves the reviewer’s concerns regarding our annotation process*
> ***
> ### **2. VLM-generated facial and environmental description (W2, Q2)**
> **While we respect the reviewer’s concern about using GPT-4o as a strong deepfake detector, it unfortunately reflects a misreading of our approach.** Our approach does not use MLLMs for capturing inconsistency that occurs in deepfakes. We resolve this curiously overlooked by clarifying our usage context, citing supporting literature, and providing both internal verification and quantitative evidence.
> * **Clarification of Our Approach**
> If we were employing the MLLM for forensic analysis (e.g., extracting forensic cues), a rigorous verification process might be necessary, as seen in studies [2, 3, 4, 5]. However, our research uses MLLM (InternVL 2.5) only to extract **global semantic information** (facial and environmental attributes). As the reviewer noted, **this is precisely what MLLMs excel at**. Aligning with this recognition, recent literature has increasingly leveraged MLLMs for semantic annotation [6,7].
> * **Related research in risk of our work**
> While specific accuracy metrics are not included, recent experimental findings [8] strongly support our approach. This study specifically investigated the **reliability of VLMs for facial attribute annotation (on the CelebA dataset)** and demonstrated that SOTA VLMs achieve **89.1% consistency** with verified human annotations. **This quantitative evidence directly refutes the concern that such annotations are 'highly unreliable'.** The study further notes that **the remaining discrepancies often stem from inherent subjectivity** (e.g., hair shade, attractive eyes, big nose) rather than hallucinations, confirming that the semantic signal provided by VLMs is robust and trustworthy for our purpose.
> * **Internal Verification & Demonstrated Effectiveness**
> To provide stronger internal verification, we manually audited randomly sampled images from each dataset and confirmed that the generated descriptions align with the corresponding visual content. We demonstrate our approach’s effectiveness quantitatively (Table 3) under a rigorous and realistic evaluation protocol, adopting seven unseen datasets. Moreover, the ablation study, as explicitly shown in Table 5, proves that both the Facial and Environmental descriptions consistently improve performance.
>
> *We trust that our rationale has been clearly conveyed, resolving the misunderstandings and encouraging a fresh assessment of our work. Accordingly, we have incorporated these into the revised manuscript in Section 3.2*.
> ***
> ### **3. Reasonable and Qualitative analysis of CLAM (W3, Q3)**
> **We explicitly present the rationale for introducing CLAM** with references to prior comprehensive research [9, 10] in Section 4.1. These studies emphasize that low-level features are helpful for distinguishing synthetic images. Consistent with these findings, we also quantitatively demonstrate that CLAM contributes to performance improvement, as clearly shown in Table 5 (+1.6% mAUC).
>
> *In response to the reviewer’s request, we additionally included t-SNE comparisons between models with and without CLAM in our revised paper (See Section 5.3). These visualizations further show that CLAM leads to more separable and structured feature representations.*

---

> ### Author Response · Authors · 2025-11-25
>
> ### **4. Spurious Correlation in Dual Contrastive Loss (Q4)**
>
> We recognize the importance of quantifying the disentanglement effect of our Dual Contrastive Loss ($\mathcal{L}_{D}$). While due to the absence of a dataset specifically designed for this analysis, we cited relevant research [11] that provides the theoretical and empirical foundation for this problem.
>
> * **Standard CLIP has spurious correlation**
> The study [11] fundamentally demonstrates that standard CLIP models suffer from **spurious correlations due to background-object biases** *(e.g., detecting polar bears is significantly easier on white backgrounds than on green ones)*. This clearly indicates that the standard CLIP objective is inherently vulnerable to such biases. **We hypothesize that this same issue remains in deepfake detection, where models may rely on context rather than forensic traces**. This vulnerability motivated us to introduce $\mathcal{L}_{D}$ in deepfake detection.
>
> * **Rationale of introducing $\mathcal{L}_{D}$**
> Our model is designed to detect synthetic images by learning forensic cues, while simultaneously aligning facial and environmental contexts through $\mathcal{L}_{D}$. **By decoupling the optimization of $\mathcal{L}_{D}$ from the main detection task**, we believe that $\mathcal{L}_{D}$ effectively mitigates spurious correlations.
>
> * **Substantial performance gain from $\mathcal{L}_{D}$**
> **Consistent with our hypothesis**, our Ablation Study (Tab. 5) empirically demonstrates the necessity of the $\mathcal{L}_{D}$ loss, which yields a substantial performance boost of +5.95% mAUC upon introduction. We acknowledge the value of a controlled counterfactual experiment and commit to pursuing it in future work should such a domain-specific dataset be designed.
>
> *We agree that the suggested experiment would provide further insights. However, we respectfully emphasize that our primary focus is on developing a robust and highly scalable deepfake detector. We leave this investigation for future work*
> ***
> ### **5.  AIGC benchmark in Table 4 (W4, Q5)**
> **As stated in Section 5.2.2, our model uses only the MMI-DD–pretrained vision encoder, which remains frozen**. We perform linear probing only, meaning that **only a classification head** is trained on SD v1.4 fake images and ImageNet real images. **This linear probing method is a fundamental concept and conventional approach when evaluating representation quality from a pretrained encoder.**
>
> Therefore, in our case, the term "fine-tuning" in the evaluation protocol refers exclusively to training **the linear classifier**, not to updating the encoder itself. For all baseline methods, we follow their original training procedures exactly as described in their papers.
>
> **Therefore, we respectfully disagree with the reviewer’s assertion of a contradiction. The results in Table 4 directly reflect the contribution of our training architecture and the MMI-DD dataset design, not any incidental benefit from the auxiliary fine-tuning data when compared to baselines such as CLIP-SVD.**
>
> ***
> ### **6. Computational resources (Q6)**
> We acknowledge that a cross-baseline comparison of training time under strictly identical environments is not feasible within the rebuttal period. However, **we would like to assert that the computational cost of $\mathcal{SD}^2$ is highly competitive due to our design.**
> * **Training time of  $\mathcal{SD}^2$**
> **By leveraging LoRA, our framework significantly enhances training efficiency.** Training was conducted on 8 NVIDIA GeForce RTX 3090 (24GB) GPUs. Under the settings of a batch size of 64 per GPU and FP16 precision, the model exhibited a peak memory usage of 21.9 GB, completing the entire training process in approximately 12 hours."
>
> * **Inference time of  $\mathcal{SD}^2$**
> We clarify that our system leverages pre-defined text prompts to provide detection scores via cosine similarity. We pre-calculate and store inference prompt vectors. Consequently, during inference, the runtime is determined solely by the forward pass through the vision encoder. Therefore, the overall inference time is comparable to the other CLIP-based methods.

---

> ### Author Response · Authors · 2025-11-25
>
> REFERENCES
> [1] DeepFaceLab. Deepfacelab. https://github.com/iperov/DeepFaceLab, 2023. Accessed: 2023-01-01.
> [2] Yue Zhang, et al. "Common sense reasoning for deepfake detection." *European conference on computer vision*. Cham: Springer Nature Switzerland, 2024.
> [3] Ziyin Zhou, et al. "AIGI-Holmes: Towards Explainable and Generalizable AI-Generated Image Detection via Multimodal Large Language Models." In *Proceedings of the IEEE/CVF International Conference on Computer Vision (ICCV)*, 2025.
> [4] Siwei Wen, et al. "Spot the fake: Large multimodal model-based synthetic image detection with artifact explanation." In *Proceedings of the 39th Conference on Neural Information Processing Systems (NeurIPS)*, 2025.
> [5] Hengrui Kang, et al. "LEGION: Learning to ground and explain for synthetic image detection."  In *Proceedings of the IEEE/CVF International Conference on Computer Vision (ICCV)*, 2025.
> [6] Xudong Wang, et al. "TF-FAS: twofold-element fine-grained semantic guidance for generalizable face anti-spoofing." *European Conference on Computer Vision*. Cham: Springer Nature Switzerland, 2024.
> [7] Sachit Menon and Carl Vondrick. "Visual classification via description from large language models." In *The Eleventh International Conference on Learning Representations (ICLR)*, 2023.
> [8] Haoming Lu and Feifei Zhong. "Can Vision-Language Models Replace Human Annotators: A Case Study with CelebA Dataset." In *The Thirty-Eighth Annual Conference on Neural Information Processing Systems Workshop (EvalEval)*, 2024.
> [9] Christos Koutlis and Symeon Papadopoulos. "Leveraging representations from intermediate encoder-blocks for synthetic image detection." *European Conference on Computer Vision*. Cham: Springer Nature Switzerland, 2024.
> [10] Shilin Yan, et al. "A sanity check for ai-generated image detection." In *The Thirteenth International Conference on Learning Representations (ICLR)*, 2025.
> [11] Qizhou Wang, et al. "A sober look at the robustness of clips to spurious features." In *The 38th Conference on Neural Information Processing Systems (NeurIPS)*, 2024.
> ***
> > We appreciate the reviewer's effort to engage with the paper. Regrettably, the majority of concerns appear to stem from a misreading and overlooking of our settings. We trust that our revisions have comprehensively resolved these misunderstandings, and we expect the reviewer to reconsider a score that, in its current form, is not substantively defensible and does not accurately reflect the substance or contributions of this work.

---

> ### Comment · Area_Chair_xFnc · 2025-11-28
>
> Dear Reviewers,
>
> Thank you for your time and thoughtful feedback on this manuscript.
>
> The authors have now submitted their rebuttal. If you haven’t already, we kindly ask you to review their responses and consider whether your concerns have been adequately addressed.
>
> Best regards,
>
> AC

---

> > ### Author Response · Authors · 2025-11-29
> >
> > We respectfully point out to the AC that the "Strong Reject (0)" rating stems from fundamental misunderstandings of the established methodology and our specific implementation. It stands in stark contrast to the consensus of other reviewers (6, 6, 6). Specifically, the reviewer made:
> >
> > **1. A factual error (W1 / Q1):** Misinterpreting an objective, generator-level annotation as a subjective task.
> > **2. A conceptual error (W2/Q2):** Basing a core criticism on the false premise that we used  MLLMs for forensics cues, when our paper clearly states we used them for their proven strength: global semantic annotation, supported by recent literature [8].
> > **3. A technical error (W4/Q5):** Confusing the standard Linear Probing protocol with full fine-tuning.
> >
> > Given that the 0 rating is predicated on clear and demonstrable technical misreadings of our methodology, it is not substantively defensible and stands in stark contrast to the three highly confident acceptance scores (6, 6, 6).

---

### Meta-Review · Area_Chair_8Qpm · 2025-12-24

**Summary:**

This paper studies large-scale deepfake detection and proposes a unified visual–language framework (SD²) together with a large multi-modal dataset (MMI-DD) that consolidates multiple existing benchmarks with fine-grained type annotations and VLM-generated descriptions. Reviewers generally agree that the scale and diversity of the dataset are impressive, and that the proposed framework achieves strong empirical performance across both intra-domain and cross-domain benchmarks, substantially outperforming existing CNN- and CLIP-based baselines. After considering the rebuttal and post-discussion feedback, several concerns remain regarding attribution of performance gains, annotation reliability, and the degree of methodological novelty beyond large-scale data integration. However, the rebuttal has clarified key misunderstandings about the annotation protocol, the role of VLM-generated descriptions, and the evaluation settings, alleviating the most critical technical objections. Overall, while some aspects of the method would benefit from stronger controlled analyses, the paper presents a significant data-centric contribution and a well-engineered, scalable framework that is likely to be impactful and of interest to the ICLR community.

**Reviewer Concerns:**

Reviewer MC2Z: Concerns remain about annotation reliability and whether VLM-generated descriptions introduce noise or bias, though the rebuttal clarifies that these descriptions are used for global semantics rather than forensic cue extraction.

Reviewer 6zHj: The framework is technically sound and empirically strong, but further controlled experiments would help disentangle the contributions of dataset scale, data curation, and architectural components.

Reviewer uz38: While the rebuttal resolves several implementation details, questions about dataset bias, spurious correlations, and the incremental novelty over prior CLIP-based adaptations remain open.

Reviewer cYdy: The experimental results are convincing, but additional analysis would strengthen the understanding of how individual losses and modules contribute beyond large-scale training.

**Reviewer Scores:**

Reviewer MC2Z: Likely no change.

Reviewer 6zHj: Likely no change.

Reviewer uz38: Likely no change or slight increase.

Reviewer cYdy: Likely no change.

---

### Decision · Program_Chairs · 2026-01-26

Accept (Poster)